**SOFTWARE**

# Evaluating topography of mutational signatures with SigProfilerTopography

Burçak Otlu[1,2,3,4] and Ludmil B. Alexandrov[1,2,3,5*]

*Correspondence:
L2alexandrov@health.ucsd.edu

[1] Department of Cellular and Molecular Medicine, UC San Diego, La Jolla, CA 92093, USA
[2] Department of Bioengineering, UC San Diego, La Jolla, CA 92093, USA
[3] Moores Cancer Center, UC San Diego, La Jolla, CA 92037, USA
[4] Department of Health Informatics, Graduate School of Informatics, Middle East Technical University, Ankara 06800, Turkey
[5] Sanford Stem Cell Institute, University of California San Diego, La Jolla, CA 92037, USA

## Abstract

The mutations found in a cancer genome are shaped by diverse processes, each displaying a characteristic mutational signature that may be influenced by the genome's architecture. While prior analyses have evaluated the effect of topographical genomic features on mutational signatures, there has been no computational tool that can comprehensively examine this interplay. Here, we present SigProfilerTopography, a Python package that allows evaluating the effect of chromatin organization, histone modifications, transcription factor binding, DNA replication, and DNA transcription on the activities of different mutational processes. SigProfilerTopography elucidates the unique topographical characteristics of mutational signatures, unveiling their underlying biological and molecular mechanisms.

**Keywords:** Mutational signatures, Somatic mutations, Genome topography

## Background

Somatic mutations are found across the genomic landscapes of all cancers and of all normally functioning somatic cells [1, 2]. These mutations are carved by the activities of endogenous and exogenous mutational processes with each process exhibiting a characteristic mutational pattern, termed, *mutational signature* [3–5]. Prior studies have demonstrated that mutations are not uniformly distributed across the genome and that most mutational signatures are affected by the topographical features of the human genome [6, 7]. Specifically, mutational signatures can have distinct enrichments, depletions, or periodicities in the vicinity of early and late replicating regions [8, 9], genic and intergenic regions [10, 11], nucleosomes [12, 13], dense chromatin regions [14], histone modifications [15], and transcription factor binding sites [16, 17]. Additionally, some mutational signatures also exhibit transcription strand asymmetries, replication strand asymmetries, and/or strand-coordinated mutagenesis [18, 19].

While there is a plethora of bioinformatics tools for analysis of mutational signatures [20–32], to the best of our knowledge, only MutationalPatterns [22], TensorSignatures [31], and Mutalisk [32] consider a subset of topographical features as part

of their analyses. Mutalisk performs certain topographical analysis for all somatic mutations in a sample, but it does not consider the activities of different mutational signatures which can have their own distinct topographical behaviors [32]. MutationalPatterns allows comparing the mutational patterns between different regions of the human genome and it can be used for testing enrichments or depletions using Poisson tests [22]. However, the tool does not consider the structure of the genome, the patterns of different mutational signatures, and the activities of these signatures when performing statistical comparisons. In addition, a subset of topography features has also been considered in extracting de novo composite mutational signatures by TensorSignatures [31], although prior benchmarking revealed sub-optimal performance when compared to traditional tools for analysis of mutational signatures [33]. In addition, the topography capabilities of all three tools are generally focused on single base substitutions and they do not support evaluating genome topography with user-provided experimental assays such as assay for transposase-accessible chromatin with sequencing (ATAC-Seq), replication sequencing (Repli-Seq), micrococcal nuclease sequencing (MNase-Seq), and chromatin immunoprecipitation sequencing (ChIP-Seq).

In this paper, we present SigProfilerTopography—an automated bioinformatics tool for comprehensive profiling of the topography of mutational signatures of all small mutational events, including single base substitutions (SBSs), doublet base substitutions (DBSs), and small insertions and deletions (IDs). The tool supports examining data from a wide variety of user-provided experimental assays and can reveal dependencies between mutational signatures and chromatin accessibility, nucleosome occupancy, histone modifications, transcription factor binding sites, replication timing, transcription strand asymmetries, replication strand asymmetries, strand-coordinated mutagenesis, and other genome topography features. Moreover, SigProfilerTopography statistically compares all results with simulation data that accounts for the genome structure as well as the strengths and patterns of all operative mutational signatures within an examined sample. SigProfilerTopography is freely available for download from https://github.com/AlexandrovLab/SigProfilerTopography with an extensive documentation at https://osf.io/5unby/wiki/home/. The implementation of the tool (Fig. 1) and exemplars of applying SigProfilerTopography to 552 previously generated whole-genome sequenced esophageal squamous cell carcinomas (ESCCs) [34] are present in this manuscript.

(See figure on next page.)
**Fig. 1** Overview of SigProfilerTopography. **A** SigProfilerTopography takes topography feature files and somatic mutations in VCF, MAF, and text formats as input. **B** SigProfilerTopography simulates real somatic mutations *n* times using SigProfilerSimulator while maintaining a preset mutational channel resolution. **C** Real and simulated mutations are annotated with mutational channel information using SigProfilerMatrixGenerator. **D** Real and simulated mutations are probabilistically attributed to different mutational signatures using SigProfilerAssignment. Alternatively, users can provide input matrices with signatures and their respective activities. **E** False-positive rates are controlled for all somatic mutations by selecting mutations highly likely to be generated by a specific mutational signature (average probability of $\geq 90\%$ by default). For all downstream analysis, statistical comparisons are performed between real and simulated somatic mutations that are highly likely to be generated by a specific mutational signature. **F** Example outputs from occupancy, strand asymmetry, replication timing, propensity of somatic mutations near topography features, and strand-coordinated mutagenesis analyses are displayed

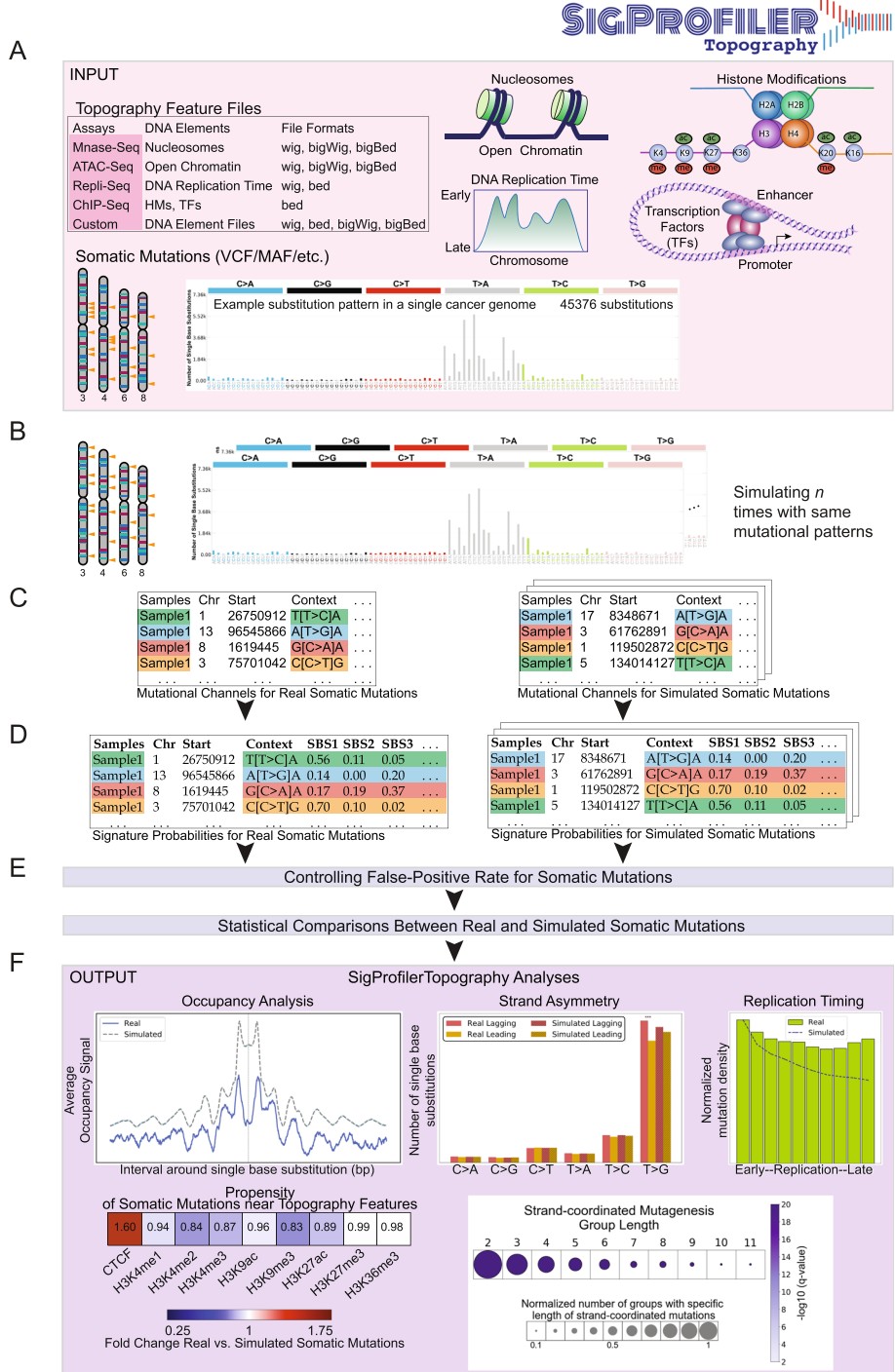

**Fig. 1** (See legend on previous page.)

## Results

### Implementation and computational workflow

As input, SigProfilerTopography requires a set of topographical features of interest and a compendium of somatic mutations from a set of samples (Fig. 1A). Topographical

features can be derived from different genomic assays (e.g., ATAC-seq, Repli-seq, MNase-seq, ChIP-seq) and these features can be inputted in a number of standard file formats, including wig, bigWig, bed, or bigBed. SigProfilerTopography's support for multiple input formats allows for topographical features to be directly downloaded from the Encyclopedia of DNA Elements (ENCODE) [35] or these features can be provided from user-generated experimental datasets. Similarly, SigProfilerTopography can examine somatic mutations using commonly supported file formats, including Variant Call Format (VCF) and Mutation Annotation Format (MAF). By default, SigProfilerTopography utilizes SigProfilerAssignment [36] to attribute the activities of known reference mutational signatures from the Catalogue Of Somatic Mutations In Cancer (COSMIC) database [37] to each examined sample. Alternatively, if another tool for assigning mutational signatures is preferred, users can provide two additional input matrices that include the patterns and activities of all operative mutational signatures in the examined samples. In either case, SigProfilerTopography will utilize the signatures' patterns and their activities to derive the probability for each mutational signature to generate each type of somatic mutation [33].

After processing the input data, SigProfilerTopography simulates all somatic mutations in each sample *n* times (default of $n = 100$) using SigProfilerSimulator [38] (Fig. 1B). SigProfilerSimulator is a computational tool that generates realistic mutational profiles as null hypotheses for statistical testing in cancer genomics. It redistributes mutations across the genome or within specific regions while maintaining key features such as mutational burden across chromosomes (or predefined regions) and local sequence context (e.g., trinucleotide or pentanucleotide context) [38]. By incorporating SigProfilerSimulator, SigProfilerTopography generates biologically constrained simulations to establish an empirical null distribution, enabling the detection of significant mutational enrichments or depletions beyond background noise. By default, the simulation resolution ensures that the total number of mutations per chromosome is preserved, along with the trinucleotide context of each somatic mutation, which includes the mutated base and its immediate 5′ and 3′ neighboring bases. After performing the simulations, both real and simulated somatic mutations are categorized in their appropriate mutation types (Fig. 1C) and a mutational signature is probabilistically attributed to each somatic mutation (Fig. 1D). SigProfilerTopography controls the false-discovery rate and, by default, only statistically compares mutations with an average of 90% probability of being caused by a specific mutational signature (Fig. 1E). Lastly, the tool outputs a variety of results allowing to distinguish differences in the topographical distribution of real somatic mutations when compared to the distribution of simulated mutations. Example analyses include evaluations of occupancy, strand asymmetries, replication timing, enrichments/depletions, and strand-coordinated mutagenesis (Fig. 1F).

### Analysis of feature occupancy

For a given topographical feature of interest, the tool evaluates the signal for detecting this feature in the vicinity, default of $\pm 1$ kilobase (kb) flanking regions, of each examined somatic mutation (Fig. 2A). The signal is aggregated for each flanking genomic position across all somatic mutations and averaged based on all available data (Fig. 2A). In the rare case of no signal being found for a specific flanking location across all mutations,

the average signal is reported as zero. Occupancy analysis is jointly performed for both real and simulated somatic mutations, thus, allowing statistical comparisons of the flanking patterns and any enrichments/depletions between real and synthetic mutations. Occupancy analysis is commonly performed to evaluate the effect of nucleosome occupancy, open chromatin, transcription factor binding sites, and histone modifications on the accumulation of somatic mutations from specific mutational signatures [6, 18].

To illustrate SigProfilerTopography's capabilities for occupancy analysis, we examined the effect of nucleosome occupancy (measured by MNase-seq data) and binding of CTCF (based on ChIP-seq data), a key regulator of chromatin architecture, on mutational signatures SBS17b and ID2 in the ESCC cohort. Signature SBS17b has a generally unknown etiology with prior studies reporting associations with damage from reactive oxygen species [39] and possible exposure to 5-fluorouracil chemotherapy [40]. Mutations due to SBS17b exhibited periodicity with a period of approximately 190 basepairs reflecting the nucleosome positions (Fig. 2B). This periodicity has been previously attributed to high damage [41] and less repair at nucleosome positions [42]. Additionally, SBS17b substitutions were highly enriched at CTCF binding sites, which is strikingly different when compared to expected by chance from the simulated substitutions (Fig. 2B). Signature ID2 has been previously attributed to slippage during DNA replication of the DNA template strand and this signature can be highly enriched in cells that are mismatch repair deficient [5]. Mutations due to ID2 were preferentially depleted at nucleosome-occupied regions (Fig. 2C) while significantly enriched at CTCF binding sites (Fig. 2C).

In addition to evaluating the patterns in the vicinity of a topographical feature, SigProfilerTopography allows summarizing the different enrichments and depletions of topographical features in the vicinity of somatic mutations when compared to synthetic mutations. Specifically, the tool performs a statistical test to evaluate whether the

(See figure on next page.)

**Fig. 2** Evaluating occupancy of topographical features. **A** Conceptual and simplified depiction of SigProfilerTopography's occupancy analysis, where x-axes correspond to ±1 kilobase (kb) from the genomic positions of real and simulated mutations. Colored boxes reflect the experimental signal detected for a specific genomic location while white boxes correspond to no experimental signal. **B** Nucleosome occupancy analysis (left panel) and CTCF occupancy analysis (right panel) exemplars for substitution signature SBS17b. **C** Nucleosome occupancy analysis (left panel) and CTCF occupancy analysis (right panel) exemplars for indel signature ID2. In **B** and **C**, solid lines and dashed lines display the average topography feature's signal (y-axes) along a 2-kilobase window (x-axes) centered at the somatic mutation locations for real and simulated mutations, respectively. The mutation location is annotated in the middle of each plot and denoted as 0. The 2-kilobase window encompasses 1000 base-pairs 5′ adjacent to each mutation as well as 1000 base-pairs 3′ adjacent to each mutation. **D** The heatmap shows enrichments and depletions of ESCC signatures within CTCF transcription factor binding sites, histone modifications, and nucleosomes. These values are calculated based on a genome-wide null hypothesis that considers mutation rates across the entire genome. **E** The heatmaps depict the enrichment and depletion of ESCC signatures within CTCF transcription factor binding sites and H3K4me1 histone modification regions. Here, enrichments and depletions are assessed relative to a null hypothesis that incorporates somatic mutation rates within the regions of interest and their adjacent 10-kilobase flanking regions on both the 5′ and 3′ ends. In **D** and **E**, red colors correspond to enrichments of real mutations and blue colors correspond to depletions of real mutations when compared to simulated data. The intensities of the red and blue colors reflect the degree of enrichments or depletions based on the average fold change. White color boxes with no annotation correspond to insufficient data for performing statistical comparisons. Statistically significant enrichments and depletions are annotated with * (*q* value ≤ 0.05)

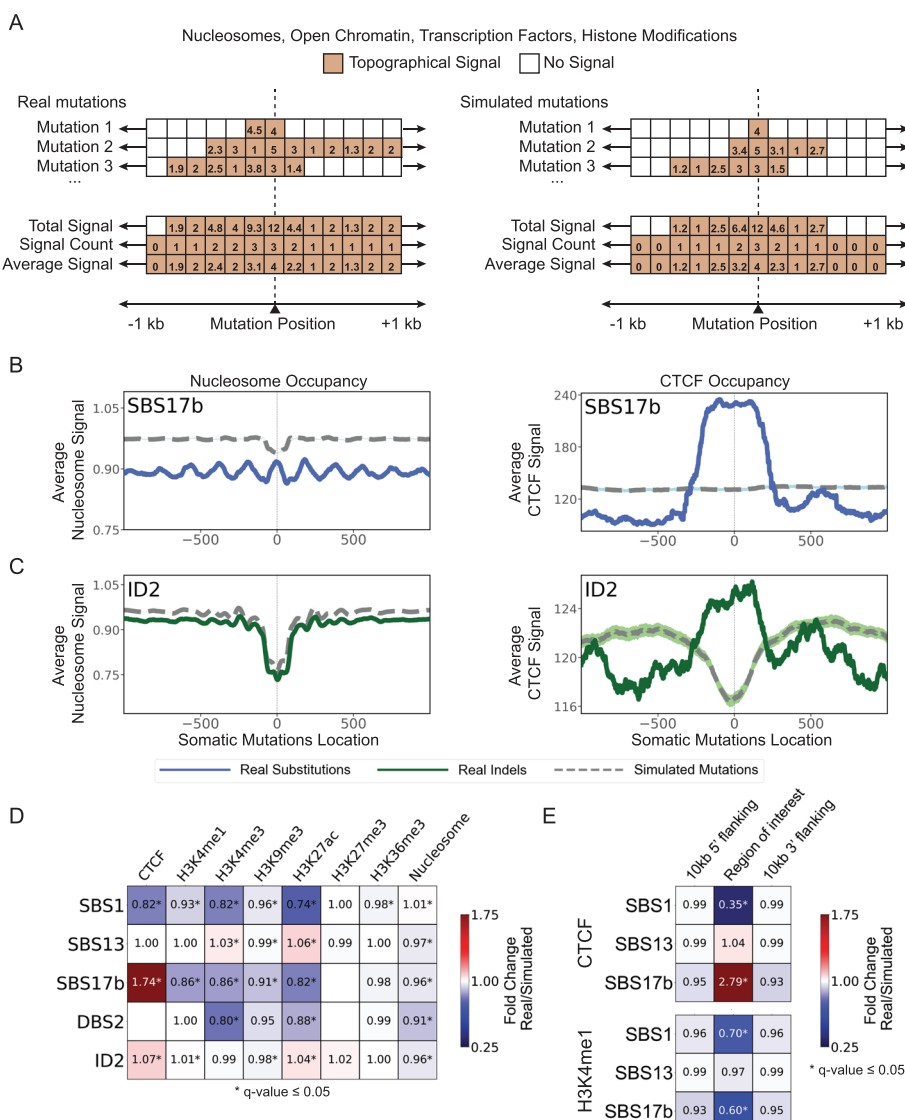

**Fig. 2** (See legend on previous page.)

topographical signal is enriched, depleted, or as expected based on the simulated data. Applying SigProfilerTopography to 8 topographical features and 5 mutational signatures in the ESCC cohort reveals that mutational signatures can be distinctly affected by each topographical feature. For example, SBS17b is enriched in CTCF binding sites and depleted at histone marks (Fig. 2D). This depletion is especially profound at H3K4me1 and H3K27ac, both of which delineate enhancer regulatory regions [43, 44].

The results presented in Fig. 2D are based on a genome-wide null hypothesis that simulates all mutations across the entire genome. However, some of the reported findings, such as the depletion of SBS17b in H3K4me1-marked enhancers, could be an indirect effect of a general depletion of SBS17b in open chromatin. To assess this possibility, SigProfilerTopography can utilize SigProfilerSimulator [38] which enables testing of different and targeted null hypotheses, allowing for a more refined analysis of regional

mutational rates and patterns. To demonstrate this flexibility, we re-evaluated the distribution of SBS1, SBS13, and SBS17b within CTCF-bound regions and H3K4me1-marked enhancers using a local background mutation rate derived by simulating somatic mutations only within the regions of interest and their 10-kilobase flanking regions. This approach ensures that the null hypothesis is derived from somatic mutations within the local region rather than the entire genome.

The results revealed that using a local background strengthened the observed effect sizes for SBS1 and SBS17b (Fig. 2E). Specifically, the depletion of SBS1 in CTCF and H3K4me1-marked regions was more pronounced, shifting from 0.82-fold and 0.93-fold reductions under the genome-wide null hypothesis to 0.35-fold and 0.70-fold reductions under the local background model. Similarly, for SBS17b, the enrichment in CTCF regions increased from 1.74-fold to 2.79-fold, while the depletion in H3K4me1-marked enhancers became more pronounced, decreasing from 0.86-fold to 0.60-fold. Notably, no significant differences were detected for the flanking regions, reinforcing that these effects are specific to the regions of interest. In contrast, SBS13 showed no significant differences regardless of the background mutation rate used (Fig. 2E). These results highlight the importance of considering local genomic context when analyzing regional mutational patterns and underscore the adaptability of SigProfilerTopography in testing different null hypotheses to refine biological interpretations.

### Evaluating replication timing

Cells replicate their DNA following a predefined replication timing program [45–47]. DNA replication begins simultaneously at multiple origins of replication and propagates bidirectionally on both strands. Chromosomal regions close to the origin of replication will replicate early, whereas regions that are far from the origin will replicate late. SigProfilerTopography can infer early and late replicating regions based on Repli-seq assay (Fig. 3A) as well as replication strand asymmetries (Fig. 3B). For the

(See figure on next page.)

**Fig. 3** Examining the effect of replication timing and replication strands. **A** DNA replication starts at multiple origins simultaneously. Genomic regions close to replication initiation zones are replicated early, whereas genomic regions close to replication termination zones are replicated late. **B** Replicational strand classification. DNA replication starts at multiple origins of replication at the same time bidirectionally at both strands. Having the same direction for DNA synthesis and replication fork migration enables continuous DNA synthesis, which results in regions on the leading strand, whereas opposite directions of DNA synthesis and replication fork cause discontinuous DNA synthesis in small fragments, termed, Okazaki fragments, on the lagging strand. **C** Mutational profile of APOBEC-associated substitution signature SBS2 using the conventional 96 mutation type classification (top panel). Replication timing analysis for substitution signature SBS2 (bottom left panel). The x-axis depicts the 10 bins from early to late replication regions, while the y-axis shows the normalized mutation density for each replication domain. The dashed line reflects the behavior of simulated mutations. Replicational strand asymmetry for substitution signature SBS2 (bottom right panel). In replication strand asymmetry figure, x-axis displays six substitution subtypes based on the mutated pyrimidine base: C > A, C > G, C > T, T > A, T > C, and T > G. Mutations were oriented by the pyrimidine base of the reference Watson–Crick base-pair and classified as ones occurring on the leading or lagging strand. The y-axis represents the number of mutations on leading and lagging strands. Real and simulated mutations are shown in bar plots and shaded bar plots, respectively. Statistically significant replication strand asymmetries are depicted with * (*q* value ≤ 0.05). **D** Mutational profile of substitution signature SBS17b using the conventional 96 mutation type classification (top panel). Replication timing analysis for substitution signature SBS17b (bottom left panel). Replicational strand asymmetry for substitution signature SBS17b (bottom right panel). Data in all panels are presented in a format analogous to that in **C**

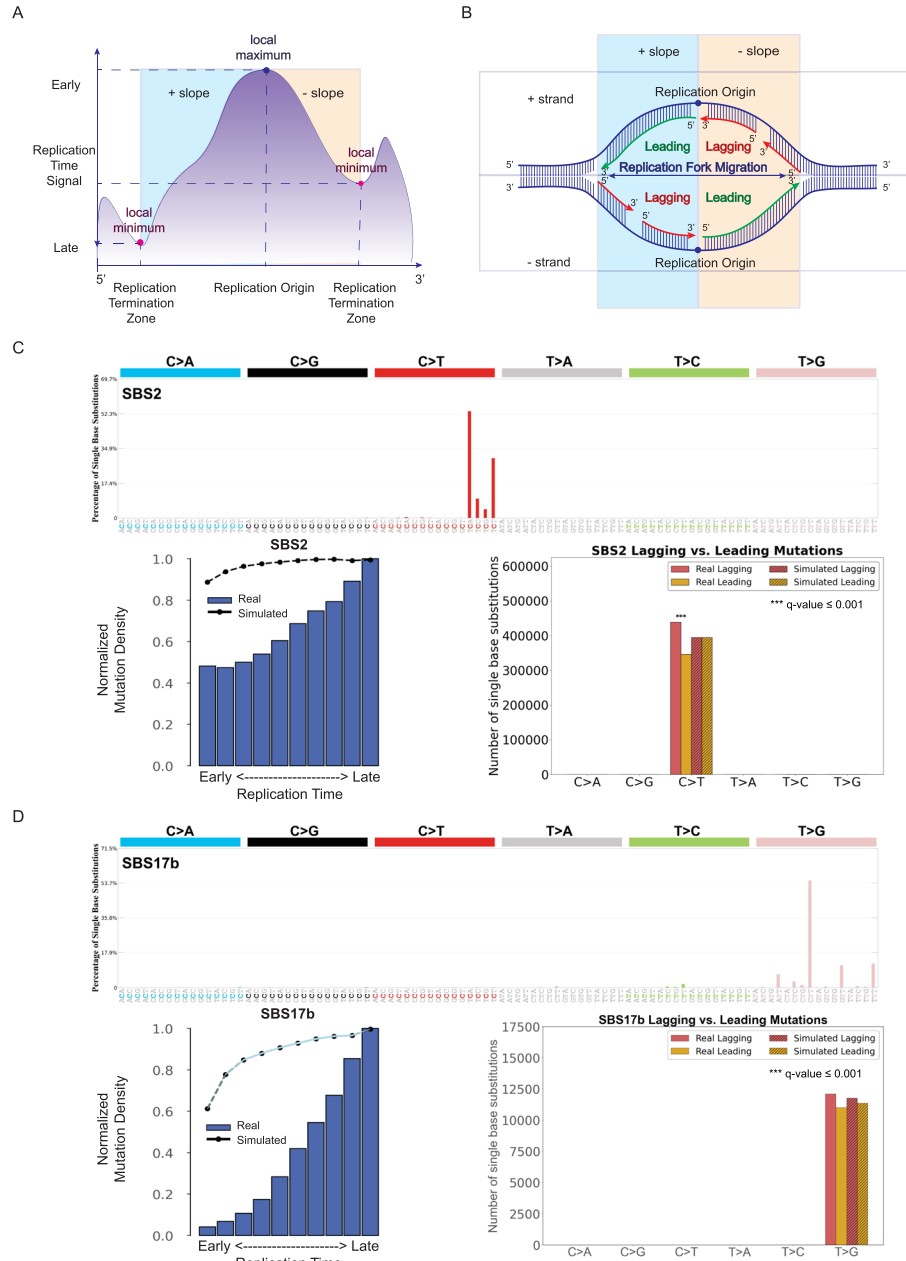

**Fig. 3** (See legend on previous page.)

analysis of early and late replicating regions, the tool leverages the Repli-seq signal, where higher values indicate earlier replication timing [48, 49]. To define replication domains, SigProfilerTopography identifies local minima and maxima in the provided signal by smoothing the weighted average data and applying wavelet-based transformation. This approach delineates regions with high signal values, representing domains of early replication where DNA synthesis initiates earlier in S-phase or in a greater proportion of cells. Local maxima and local minima in the wavelet-smoothed signal data correspond to replication initiation zones (peaks) and replication termination zones (valleys), respectively (Fig. 3A).

SigProfilerTopography uses wavelet-smoothed signal data in replication timing analysis and, additionally, peaks and valleys data in replicational strand asymmetry analysis. After sorting the replication time signals into descending order from early to late, the tool splits the signal into deciles, with each decile containing 10% of the replication time signals. To demonstrate SigProfilerTopography's capabilities for replication timing analysis, we evaluated the effect of replication timing in the ESCC cohort on signature SBS2, a mutational signature previously attributed to the activity of the APOBEC family of deaminases [34], and signature SBS17b. Similar to prior reports [6, 18], SBS2 and SBS17b both exhibited an increasing normalized mutation density from early to late replicating regions (Fig. 3C–D).

### Examining replication strand asymmetries

In eukaryotic cells, DNA replication is initiated around multiple replication origins, from where it proceeds in both directions on both strands (Fig. 3B). The strand where the direction of DNA synthesis and growing replication fork are the same is replicated continuously and it is termed leading strand. Conversely, when the direction of DNA polymerase and the growing replication fork are opposite, then that strand (termed, lagging strand) is replicated discontinuously in short Okazaki fragments [50]. Imbalance between DNA damage and DNA repair may lead to mutations from the same type to be enriched on the leading or lagging strands.

Using data from an Repli-seq assay, SigProfilerTopography can annotate mutations as ones occurring on the leading or lagging strand by orienting them by the pyrimidine base of the reference Watson–Crick base-pair. Applying SigProfilerTopography to the mutations attributed to the APOBEC-associated signature SBS2 in the ESCC cohort reveals an enrichment of mutations on the lagging strand when compared to simulated data (Fig. 3C). This result is consistent with prior reports of APOBEC deaminases targeting single-stranded DNA during replication [51]. Conversely, SBS17b showed no statistically significant enrichment or depletion of mutations on the leading or lagging strand compared to simulated data (Fig. 3D).

### Examining transcription strand asymmetries

In addition to evaluating the effect of replication on the accumulation of mutational signatures (Fig. 3), SigProfilerTopography also allows examining the impact of transcription on somatic mutagenesis. Specifically, the tool annotates each mutation as either genic or intergenic, where genic mutations are within the genomic regions of well-annotated protein coding genes and intergenic mutations are outside these regions (Fig. 4A). Moreover, somatic mutations within well-annotated protein coding genes are further subclassified based on the pyrimidine base of the reference Watson–Crick base-pair resulting into two additional subclasses: un-transcribed mutations and transcribed mutations (Fig. 4A). This subclassification allows measuring transcription strand asymmetries due to either transcription-coupled DNA repair [52, 53] or transcription-coupled DNA damage [19]. Applying SigProfilerTopography to the somatic mutations due to SBS16, a mutational signature previously associated with alcohol consumption [54], revealed both accumulation of higher number of T > C mutations on the transcribed strand as well as an enrichment of mutations within genic regions (Fig. 4B). This topographical behavior

of signature SBS16 has been previously attributed to the role of transcription-coupled damage in actively transcribed genes [19, 55]. Conversely, analysis of signature SBS17b revealed no statistically significant asymmetry between transcribed and un-transcribed strands but showed an enrichment of mutations in intergenic regions (Fig. 4C).

### Mapping strand-coordinated mutagenesis

Prior studies have shown that strand-coordinated mutations are commonly observed, for example, due to damage on single-stranded DNA, and can form hypermutable genomic regions [56, 57]. Strand-coordinated analysis provides an additional layer of insight beyond transcription and replication strand asymmetry by identifying mutations that occur on the same DNA strand as part of a single mutational event [58, 59]. Unlike transcription and replication asymmetries, which summarize multiple independent events across the genome, strand-coordinated processivity highlights localized mutation patterns driven by specific mutagenic mechanisms. For instance, APOBEC enzymes induce strand-coordinated mutations through a single catalytic event, which can be independent of transcription or replication directionality [60, 61]. Similarly, processes such as double-strand breaks and localized hypermutation can result in strand-specific mutation accumulation [58, 59]. SigProfilerTopography allows performing analysis of strand-coordinated mutagenesis by identifying groups of consecutive mutated single base substitutions, attributed to the same mutational signatures, with no more than 10 kb distance between any two mutations. Mutations are oriented by the reference base of the Watson–Crick base-pair to ensure that they are occurring on the same strand, e.g., consecutive C > A mutations attributed to a single mutational signature. Groups of varying lengths are pooled across all samples for each mutational signature. Same procedure is repeated for simulated mutations to assess the statistical significance of the observed number of strand-coordinated mutagenesis groups with expected list of number of strand-coordinated mutagenesis groups for each group length (Fig. 5A–C).

Applying SigProfilerTopography to all mutational signatures operative in the 552 whole-genome sequenced samples revealed statistically significant strand-coordinated mutagenesis for multiple signatures (Fig. 5D). The APOBEC-attributed signatures SBS2 and SBS13 exhibited groups of up to 11 consecutive mutations likely due to APOBEC-induced *kataegis* [59, 62]. Interestingly, the flat signatures SBS5

(See figure on next page.)

**Fig. 4** Assessing the impact of the transcriptional machinery. **A** Somatic mutations within protein coding genes are oriented by the pyrimidine base of the reference Watson–Crick base-pair and classified as ones being on the transcribed or un-transcribed strand. Somatic mutations outside protein coding genes are classified as ones in intergenic region. **B** Mutational profile of substitution signature SBS16 using the conventional 96 mutation type classification (top panel). Exemplar transcriptional strand asymmetry analysis for substitution signature SBS16 (bottom left panel). Exemplar genic versus intergenic regions analyses for substitution signature SBS16 (bottom right panel). X-axes in the bottom panels display six substitution subtypes based on the mutated pyrimidine base: C > A, C > G, C > T, T > A, T > C, and T > G, and the y-axes represent the number of mutations both for real and simulated mutations on transcribed and un-transcribed strands or genetic and intergenic regions. Simulated mutations are shown in shaded bar plots. **C** Mutational profile of substitution signature SBS17b using the conventional 96 mutation type classification (top panel). Exemplar transcriptional strand asymmetry analysis for substitution signature SBS17b (bottom left panel). Exemplar genic versus intergenic regions analyses for substitution signature SBS17b (bottom right panel). Data in all panels are presented in a format analogous to that in **B**

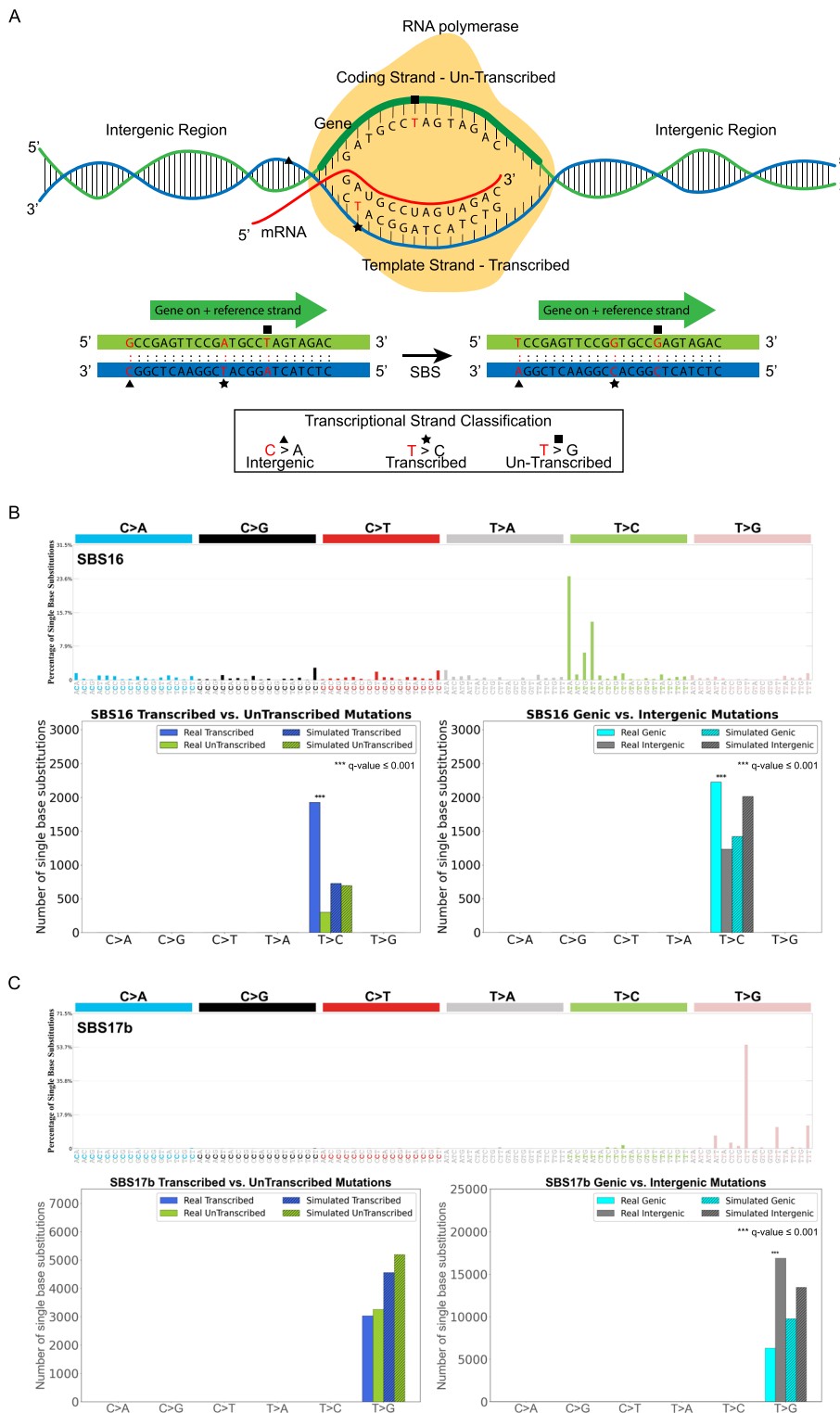

**Fig. 4** (See legend on previous page.)

and SBS40 also manifested strand-coordinated mutagenesis of varying group length. Lastly, we observed an enrichment of mutation clusters of length two across most mutational signatures (Fig. 5D), a previously described phenomenon primarily driven

by *omikli* events [63], a common subclass of clustered mutations in human cancers [58]. However, since the effective mutable genome size may be smaller than the genomic portion used for simulations [64], some of this enrichment could also stem from an overrepresentation in the observed data.

## Discussion

SigProfilerTopography is an open-source Python package that allows understanding the interplay between somatic mutagenesis and the structural and topographical features of a genome. The tool can reveal mutational signature-specific tendencies associated with chromatin organization, histone modifications, and transcription factor binding as well as ones affected by cellular processes such as DNA replication and transcription. As we illustrated by applying the tool to 552 whole-genome sequenced ESCCs, SigProfilerTopography simultaneously examines real somatic mutations and simulated mutations, compares their tendencies, and then elucidates the statistically significant differences for each structural and topographical feature of interest. The tool also seamlessly integrates with other SigProfiler tools and leverages them for parts of its computational workflow, including classification of somatic mutations using SigProfilerMatrixGenerator [65], simulating realistic background mutations with SigProfilerSimulator [38], and assigning mutational signatures to each somatic mutation using SigProfilerAssignment [36]. Notably, applying SigProfilerTopography to other cancer types produces results identical with our previously reported pan-cancer topography analyses [6].

Beyond its core functionality, SigProfilerTopography offers flexibility in defining null hypotheses, allowing users to choose between genome-wide and localized mutation backgrounds for statistical comparisons. By default, simulations assume a distribution of mutations across the entire genome, but a more refined null hypothesis can be applied by restricting the background mutation rate to specific genomic regions of interest, including their flanking sequences. This flexibility is particularly relevant when analyzing mutational patterns in regulatory elements, where the influence of chromatin accessibility, transcription factor binding, and histone modifications may be confounded by broader regional mutation rates. As demonstrated in our ESCC analysis, using a localized background strengthens effect sizes and refines biological interpretations of mutational patterns in CTCF-bound regions and H3K4me1-marked enhancers.

(See figure on next page.)
**Fig. 5** Mapping strand-coordinated mutagenesis. **A** Three simplified exemplar samples illustrating consecutive single base substitutions occurring on the same DNA strand due to specific mutational signatures. For example, three consecutive C > T mutations on the same strand generated by SBS5 within sample 1 result in one strand-coordinated mutagenesis group of length 3. **B** Summary of strand-coordinated mutagenesis groups of varying lengths for each mutational signature within each of the three examined samples from **A**. **C** Accumulation of strand-coordinated mutagenesis groups across all three examined exemplar samples from **A**. **D** Strand-coordinated mutagenesis for COSMIC substitution signatures operative in 552 ESCCs. Circle plot displays the group lengths from 2 to 11 mutations on the x-axis and the SBS mutational signatures on the y-axis. Circle size represents the number of strand-coordinated mutagenesis groups for the corresponding group length, which is normalized for each mutational signature. Circle color indicates the statistical significance of the finding with $-\log_{10}$ (*q* value), with darker colors corresponding to lower *q* values

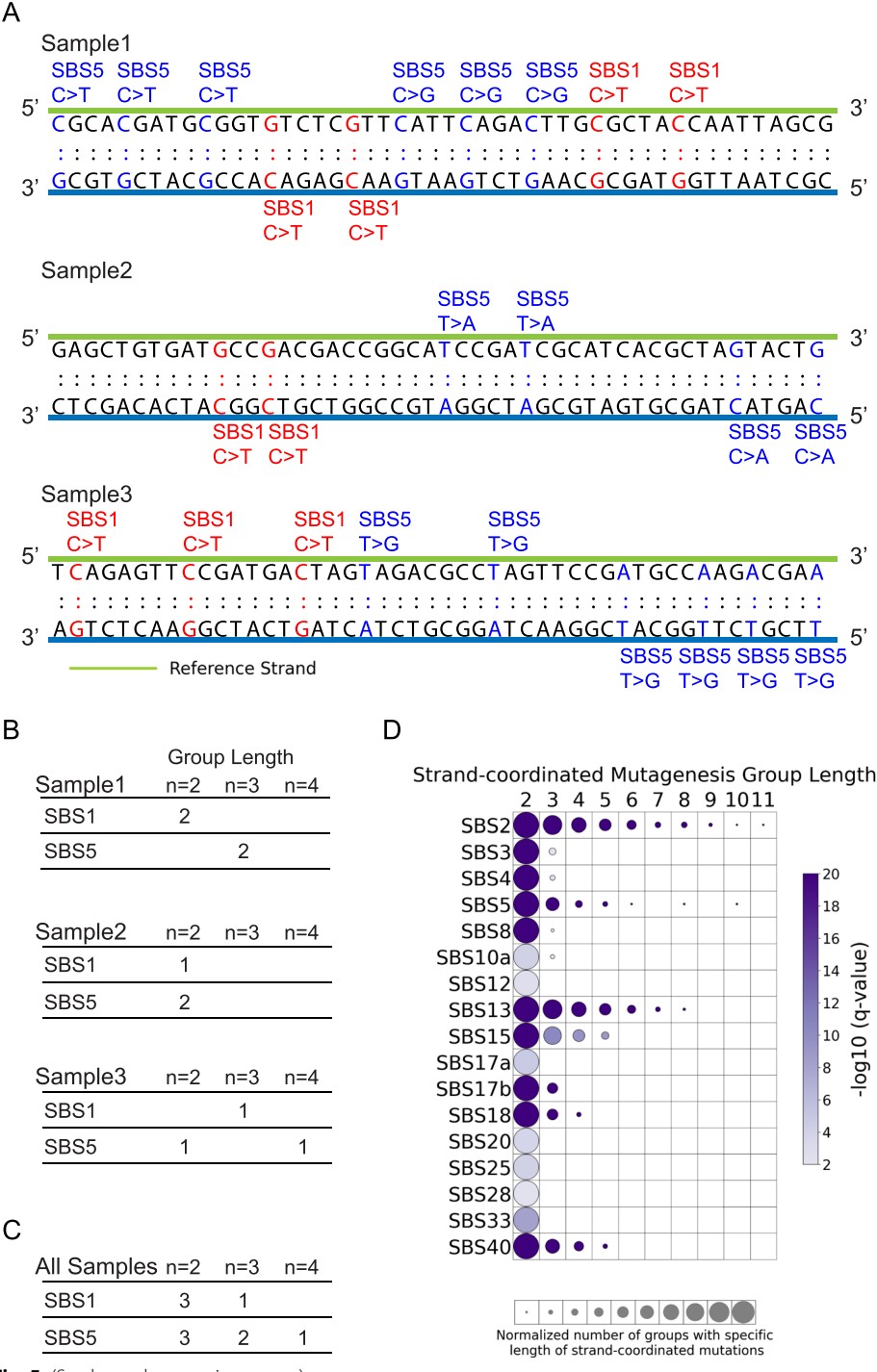

**Fig. 5** (See legend on previous page.)

One important consideration in using SigProfilerTopography is evaluating strand asymmetries for different mutational signatures. The tool currently infers strand asymmetries by considering replication timing and transcriptional strand orientation separately, yet mutational processes can be influenced by interactions between these factors [19]. Future enhancements could allow for more sophisticated models that jointly

consider replication direction, transcription direction, and strand bias, as incorporating these interactions could further refine insights into the mechanistic basis of mutation accumulation.

Additionally, SigProfilerTopography could be expanded to explore even more complex interactions between genomic features. For instance, while the current implementation assesses individual features such as replication timing, chromatin accessibility, and histone modifications independently, future iterations could integrate multi-feature interactions, such as the combined effects of replication timing, gene expression, and mutation distribution. These relationships may be particularly relevant in highly transcribed regions, where differential repair mechanisms and transcription-replication conflicts can influence mutation rates.

Our application of SigProfilerTopography to ESCC provides a compelling example of how the tool can be used to uncover biologically meaningful insights into the underlying processes driving mutagenesis. The ESCC analysis highlighted the differential behavior of SBS1, SBS13, and SBS17b across distinct genomic regions, demonstrating the importance of regional mutational landscapes in shaping mutation distributions. More broadly, the tool facilitates the systematic investigation of how mutational processes interact with the genome topography, chromatin dynamics, and cellular replication and transcription machinery, providing a powerful resource for understanding the etiology of somatic mutations in cancer.

Despite its strengths, SigProfilerTopography has certain limitations. First, the tool can only be used to explore small mutational events including single base substitutions, doublet substitutions, and small insertions and deletions. Currently, the tool does not allow exploring large mutational events [66] such as copy-number changes and structural rearrangements. Second, SigProfilerTopography can be applied only to whole-genome sequenced cancers, and it will not work on whole-exome or targeted cancer gene panel sequencing data as the algorithm requires profiling the non-coding regions of the genome. Lastly, the tool necessitates sufficient numbers of somatic mutations for the statistical analyses to be meaningful and statistically significant. We have previously shown that topographical analyses will work and can yield biologically exciting results when examining adult cancers [6], however, it is currently unclear whether some pediatric cancer genomes will have sufficient numbers of somatic mutations for examining the topography of their mutational signatures. To provide estimates of the necessary number of mutations required to detect different effect sizes, we re-examined our previously published topography analyses [6] to evaluate the minimum statistically detectable effect size and the effect size that is statistically significant in 90% of cases based on the number of mutations assigned to a mutational signature within a cancer type. In this context, statistical significance is defined as a $p$ value below 0.05 after being adjusted for multiple hypothesis testing across all signatures within a given cancer type. Our analysis indicates that no effect size could be detected for mutational signatures with fewer than 1000 mutations in a given cancer type. For signatures with 1000 to 10,000 mutations, the minimum detectable effect size was a 1.3-fold change, with 90% of detected effect sizes above a twofold change. For signatures with 10,000 to 50,000 mutations, the minimum detectable effect size was a 1.2-fold change, with 90% of detected effect sizes above a 1.7-fold change. For signatures with 50,000 to 100,000 mutations, the minimum

detectable effect size was a 1.1-fold change, with 90% of detected effect sizes above a 1.5-fold change. For signatures with more than 100,000 mutations, the minimum detectable effect size was a 1.03-fold change, with 90% of detected effect sizes above a 1.2-fold change. These findings emphasize the importance of considering mutation count when designing and interpreting topographic analyses and provide a framework for assessing the applicability of SigProfilerTopography in smaller cohorts or cancer types with low numbers of somatic mutations.

## Conclusions

SigProfilerTopography enables a thorough examination of how genome topography and genome architecture impact the accrual of somatic mutations. The tool offers a robust approach for evaluation of localized somatic mutation rates across various genomic features within a single comprehensive platform, offering a scalable solution for analyzing large datasets encompassing many thousands of cancer genomes and all types of small mutational events. Overall, SigProfilerTopography is a computational tool that provides an unprecedented opportunity for understanding the biological mechanisms and molecular processes influencing somatic mutational processes that have operated in a cancer genome.

## Methods

### Tool implementation

SigProfilerTopography is developed as a computationally efficient Python package, and it is available for installation through PyPI. The tool leverages SigProfilerAssignment for attributing mutational signatures to individual somatic mutations [36], SigProfilerSimulator for generating all simulated datasets [38], and SigProfilerMatrixGenerator for processing input data for somatic mutations [65]. SigProfilerTopography leverages SigProfilerSimulator [38] to enable highly customizable simulations by allowing users to specify different mutation resolutions and genomic regions for generating simulated mutations. By default, simulations are distributed across the entire genome while preserving the observed mutation counts for each mutation type and chromosome using the SBS-96, DBS-78, and ID-83 mutation contexts [65]. However, users can refine simulations by matching transcription strand asymmetry to real mutations while maintaining mutation type and chromosome-specific counts. This ensures that other topographic features can be analyzed without altering transcription strand asymmetry. Additionally, simulations can be restricted to specific genomic regions defined by a browser extensible data (BED) file, allowing for targeted investigations of mutational topography. SigProfilerTopography allows processing all types of small mutational events, including (i) single base substitutions, (ii) doublet base substitutions, and (iii) small insertions and deletions. The tool supports most commonly used data formats for somatic mutations: Variant Calling Format (VCF), Mutation Annotation Format (MAF), International Cancer Genome Consortium (ICGC) data format, and simple text file. SigProfilerTopography allows examining topography features in wiggle (wig), browser extensible data (bed), bigWig, and bigBed formats. The tool has been extensively tested on data from transposase-accessible chromatin with sequencing (ATAC-Seq), replication sequencing (Repli-Seq), micrococcal nuclease sequencing (MNase-Seq), and immunoprecipitation sequencing

(ChIP-Seq). As shown in the manuscript, these data can be analyzed to evaluate occupancy, transcription strand asymmetry, replication strand asymmetry, replication timing, and strand-coordinated mutagenesis. The methodology of each of these analyses is described in the subsequent sections.

### Occupancy analysis

For each real somatic mutation, SigProfilerTopography accumulates both the total signal intensity (*SUM*) and the number of signals (*COUNT*). The average signal at each base is then calculated as $SUM/COUNT$. This process is repeated for simulated mutations, allowing for statistical comparison between observed and simulated data. Confidence intervals around the average simulated occupancy signals are generated, and significance is assessed using a *z*-test. The *p* values are combined using Fisher's method and corrected for multiple testing using the Benjamini–Hochberg procedure.

### Replication timing

Replication timing signals are sorted in descending order and divided into deciles, each containing 10% of the data. The first decile represents the earliest replicating region, while the last decile corresponds to the latest replicating region. Real somatic mutations are assigned to deciles based on their overlap with replication domains, and mutation densities are calculated by normalizing against the number of valid bases (A, T, G, C) in each decile. Densities are further normalized relative to the highest mutation density. This process is repeated for simulated mutations, enabling the generation of confidence intervals around the normalized mutation densities.

### Strand asymmetry

In the Repli-seq signal data, local maxima correspond to replication initiation zones (peaks), while local minima represent replication termination zones (valleys). Peaks and valleys are sorted by genomic coordinates, and regions of at least 10 kb with a positive slope are annotated as leading strands, while negative slopes define lagging strands. To ensure robustness, the last 25 kb of replication termination zones are excluded. Fisher's exact test is used to compare the observed versus simulated mutation counts on each strand, with multiple testing correction via the Benjamini–Hochberg method. Statistically significant strand asymmetries (adjusted $p \leq 0.05$) are reported, with odds ratios above 1.10 used to identify enriched mutation biases across lagging versus leading, transcribed versus un-transcribed, and genic versus intergenic regions.

### Strand-coordinated mutagenesis

For each SBS signature and group length, the observed number of clustered mutation groups is compared to the expected number derived from 100 simulated datasets, serving as the null hypothesis. Statistical significance is assessed using *z*-tests, determining whether the observed mean differs significantly from the simulated mean. The *p* values are adjusted for multiple testing using the Benjamini–Hochberg method, and only SBS signatures and group lengths with an adjusted $p \leq 0.05$ are considered statistically significant and reported.

### Statistical testing

By default, the tool performs statistical comparisons and Benjamini–Hochberg corrections for multiple hypothesis testing using the statsmodels Python package. Where appropriate, Fisher's method is used to combine $p$ values. Adjusted $p$ values are considered statistically significant if they are below 0.05.

### Esophageal cancer dataset

A previous study [34] collected 552 esophageal squamous cell carcinomas (ESCC) including tumor and germline DNA, which were subjected to whole-genome sequencing with mean sequencing coverage of 49-fold and 26-fold, respectively. All ESCC cases underwent centralized pathology review by an expert panel of seven gastrointestinal pathologists convened by IARC/WHO. Each case was independently evaluated by at least two pathologists through blinded digital pathology assessment, and any discrepancies were adjudicated by a third independent pathologist to ensure accuracy [34]. De novo mutational signatures were extracted and decomposed into COSMIC reference signatures using SigProfilerExtractor [33]. All somatic mutations within the ESCC dataset were considered with each mutation probabilistically assigned to each of the operative mutational signatures.

## Supplementary Information

> Additional file 1. Peer review history.

### Acknowledgements
The computational development reported in this manuscript has utilized the Triton Shared Computing Cluster at the San Diego Supercomputer Center of UC San Diego. We thank Marcos Díaz-Gay for reading the draft of the manuscript and for providing feedback. We also thank Ting Yang, Mousumy Kundu, and Mark Barnes for testing the tool and for the helpful discussions. The research in this study was also supported by UC San Diego Sanford Stem Cell Institute.

### Peer review information

### Authors' contributions
BO developed the Python code and wrote the draft of the manuscript. LBA supervised the overall development of the code and writing of the manuscript. All authors read and approved the final manuscript.

### Funding
This work was supported by the US National Institute of Health grants R01ES030993-01 A1, R01ES032547-01, U01 CA290479-01, and R01 CA269919-01 to LBA as well as Cancer Research UK Grand Challenge Award C98/A24032. This work was also supported by a Packard Fellowship for Science and Engineering. The funders had no roles in study design, data collection and analysis, decision to publish, or preparation of the manuscript.

### Data availability
The somatic mutations for the 552 previously generated esophageal squamous cell carcinoma samples were retrieved from a prior submission [67]. SigProfilerTopography is freely available, distributed under the BSD-2-Clause license, and has been extensively documented. The code used in this publication can be downloaded from a research data repository [68], while version control platform for code development and collaboration is also available [69]. Additional data sharing is not applicable to this article as no datasets were generated as part of the current study.

## Declarations

### Ethics approval and consent to participate
Not applicable.

### Consent for publication
Not applicable.

**Competing interests**
LBA is a co-founder, CSO, scientific advisory member, and consultant for io9, has equity, and receives income. The terms of this arrangement have been reviewed and approved by the University of California, San Diego in accordance with its conflict of interest policies. LBA's spouse is an employee of Biotheranostics. LBA declares US provisional applications with serial numbers: 63/289,601, 63/269,033, 63/483,237, 63/366,392, 63/412,835, and 63/492,348, as well as an international patent application PCT/US2023/010679 and European patent application with application number EP25305077.7. BO declares no known competing interests or personal relationships that could have appeared to influence the work reported in this paper.

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

## 