## [Additional file 1. Peer review history. · Genome Biology]

Review history

First round of review

Reviewer 1

In this manuscript, Otlu & Alexandrov describe an extension of the SigProfiler software package called "SigProfiler Topography" that allows the annotation of somatic mutational data with "topographic" information, such as regionally variable chromatin features or replication direction. This tool is a useful addition to the already excellent SigProfiler package and will be of great use to the wider community. I only have minor comments regarding the findings presented in the manuscript.

I have few comments to make, only a handful of points relating to the biological examples chosen for demonstrating the utility of SigProfiler Topography:

- The authors state that S17b is depleted at e.g. H3K4me1. However, I wonder if this could be an indirect effect of a general depletion of S17b in open chromatin. This is a general caveat with the way the simulations are done for statistical significance testing. Using a "genome-wide" background means that it will be difficult to separate co-variables. To test whether S17b is depleted for H3Kme1, one would have to use a background that is more representative of the topology of enhancers, yet doesn't contain an enhancer (e.g. immediate flanking regions of the enhancer). While I believe this is out of the scope of this methods paper, it could be something to discuss in the (overall rather short) discussion section.
- I'm wondering whether the S17 patients in the ESCC cohort are in fact misdiagnosed adenocarcinomas? It would be really interesting to know with certainty whether squamous carcinomas do exhibit S17b, which could have implications for the aetiology of this signature
- S17b is also strongly replication-timing asymmetrical, but the authors use S2 as their example. Personally, I thought it would have been nice to show the "deep characterisation" of one signature
- Isn't the "strand-coordinated analysis" going to be highly overlapping with the findings from replication and transcription strand asymmetry? What other types of biological factors would lead to strand-specific mutation accumulation? Maybe the authors could highlight the benefits of looking at strand-coordinated patterns as opposed to replication/transcription direction
- The discussion is a bit thin. The authors could discuss that there are potentially different ways to assign signatures to strands in regions? Also, how about interactions between terms, as in replication<>gene expression<>mutation distribution, or replication direction<>transcription direction<>strand bias (as described in my earlier point)? The discussion could also make a bit more of the ESCC analysis, or at least mention that the tool can be used to really understand underlying biology of mutational processes.

Reviewer 2

Otlu and Alexandrov propose a new Python extension of the SigProfiler suite called SigProfilerTopography, which leverages SigProfilerAssignment and SigProfilerSimulator to assess association of mutation signatures with "topographic" data, including chromatin organisation, DNA replication and transcription, histone modifications and TF binding.

This generic tool is of relevance to the study of mutagenesis, especially the interrelationship between genome topography and mutability/repair. Unlike other existing tools, it

simultaneously allows to analyse SBS, DBS and indel signatures in relation to precomputed and user-defined topography data provided in standard formats.

The authors first present the tool and its computational workflow. It works from mutation calls, and signature activities to assign each mutation to a signature. It simulates N (N=100 default) random profiles with similar activities and number of mutation per chromosome to derive a null model against which to compare the observed signal.

It looks at the average observed vs. simulated topography signal in ± 1 kb of the mutation position, for all mutations and for mutations assigned to different signatures.

The authors apply this method to a large published ESCC cohort and find known relationships between signatures and topography signal.

SigProfilerTopography also looks at association of mutations from different signatures with replication timing and replication strand. For this it can leverage Repli-seq data to annotate mutations in early vs. late replicating timing regions, and annotates mutations based on the reference Watson-Crick base pair for assessing association with leading vs. lagging strand mutations. They show that SBS2 (APOBEC) is indeed seen in late-replicating regions and more in the lagging strand, which is compatible with the targeting of single-stranded DNA.

Similarly, to assess whether mutations are related to transcription, SigProfilerTopography can annotate mutations for whether they fall in well-annotated protein coding genes and whether they are on the transcribed strand. They show SBS16, related to alcohol, is seen more in genic regions and on the transcribed strand.

Finally, thanks to the simulated profiles, SigProfiler can detect enrichment in the observed data across the samples for mutations from the same process within 10kb of each other and on the same strand, akin to APOBEC-related kataegis. Interestingly, they find that SBS1, SBS5 and SBS15 also present many groups of high number of consecutive mutations on the same strand.

In conclusion, SigProfilerTopography is a useful addition to the current SigProfiler suite of tools, which integrates well with the other tools. It deals with SBS, DBS, and indels in whole-genome sequencing data, and works well provided enough mutations as input.

Altogether, I think this is a useful addition to the toolkit for assessing association between mutational signatures and genomewide annotation tracks that could influence mutation rates (here referred to as "topography"). It is definitely worth publication and I believe would be used and well cited.

However, I would have a few comments that I hope could potentially help improve the manuscript and perhaps the tool itself.

Major comments.

1) Software. The software is easy to install and the tutorials are well written.

I ran the example and only obtained figures for occupancy, not epigenomics, replication time, strand bias etc. although the logs showed that all analyses ran well. I then realised that while installing the hg19 tracks from the examples, as some of these are relatively large files, some of the files were not downloaded properly. On reinstallation, the software detected that the files were present on disk and did not redownload the corrupt files. I had to delete those files and reinstall. Even after reinstalling those files, I only obtained figures for occupancy.

Here is the code I used after downloading the input data:

```
from SigProfilerTopography import Topography as topography

topography.install_nucleosome("GRCh37")
topography.install_atac_seq("GRCh37")
topography.install_repli_seq("GRCh37")

genome = "GRCh37"
inputDir = "21BRCA_vcfs"
outputDir = "results"
jobname = "21BRCA_SPT"
numofSimulations = 5
topography.runAnalyses(genome,
inputDir,
outputDir,
jobname,
numofSimulations,
epigenomics=True,
nucleosome=True,
replication_time=True,
strand_bias=True,
processivity=True)

jobname = "21BRCA_SPT_with_probability_matrices"
numofSimulations = 5
sbs_probability_file =
"21BRCA_probabilities/COSMIC_SBS96-Decomposed_Mutation_Probabilities.txt"
dbs_probability_file =
"21BRCA_probabilities/COSMIC_DBS78-Decomposed_Mutation_Probabilities.txt"

topography.runAnalyses(genome,
inputDir,
outputDir,
jobname,
numofSimulations,
sbs_probabilities = sbs_probability_file,
dbs_probabilities = dbs_probability_file,
epigenomics=True,
nucleosome=True,
```

replication_time=True,
strand_bias=True,
processivity=True)

2) The analyses of SigProfilerTopography rely on the simulations by SigProfilerSimulator to assess significance, which is a critical part of the analyses. While SigProfilerSimulator is described in another paper, it would be helpful to briefly describe how it works in the methods of this manuscript. In particular, it is not clear to me how the mutations are distributed across the genome during the simulations, i.e. whether it is a uniform coverage only taking into account the nucleotide context and the number of mutations per chromosome or whether some of the topology tracks are also taken into account. Indeed, the simulator could in theory keep the distribution of mutations similar to the observed data in relation to topography tracks (e.g. similar number of mutation in early vs. late replicating regions) for assessing another given topography track (e.g. nucleosome or CTCF occupancy). This could allow to control for interdependencies between the topologies themselves.

3) For the mutation-cluster analysis, it seems that the FDR is often low for group length 2 across signatures. Could the authors elaborate why they think that is? Could it be that the simulations are too uniform and the effective genome size, i.e. the cancer-type-specific mutable genome portion, is shorter than the genome portion used for the simulations (see doi.org/10.1038/s41588-021-01005-8) and thus for the same number of mutations, the number of length-2 groups is higher in the observed compared to the simulations? Also related to that, does the simulator take into account "bad genomic regions" where mutation calls are difficult and thus not observed?

4) It is mentioned that it is unclear what the lower bound for the number of mutations would be to get signal and therefore, whether the tool would be applicable to pediatric cancers. Given the known number of mutations in pediatric cancers, simulations or downsampling of the ESCC mutations could help answer this question and this would also be useful information to future users with smaller cohorts.

Minor comment.

1) Figure 3B is referenced later than panels other than 3A; maybe change the order?

Reviewer 3

The authors are recognized as one of the leading research teams in the field of mutational signatures. The manuscript they have submitted appears to extend the analyses presented in their recent Cell Reports publication (2023) into a software paper for Genome Biology, with the intention of making it accessible to the broader cancer mutation research community.

Regrettably, I find that the manuscript, in its current form, does not meet the standards required for publication in the Software section of Genome Biology for the following reasons:

- The software presented lacks sufficient novelty. It would be more appropriate to offer it as part of the supplementary materials in their Cell Reports (2023) publication, where it can still be of value to the global research community.
- While the authors have provided an overview of the software on their website, the manuscript lacks the necessary depth and structure. To enhance its clarity for the readership, it should include a more detailed description of the software's architecture, implementation, and the methodologies employed (including statistical considerations).
- The authors mention three known limitations in the Discussion section. Addressing these limitations in the software's implementation would significantly elevate SigProfilerTopography to a level that could meet the criteria for publication in Genome Biology.
- Additionally, it is crucial to demonstrate whether the software can achieve a comparable performance across pan-cancer analyses, as seen in the analysis of the ESCC cancer type presented in the manuscript.

In conclusion, the manuscript, in its current form, falls short of the expectations for a software paper in Genome Biology.

Authors' response to reviewers

Reviewer #1: In this manuscript, Otlu & Alexandrov describe an extension of the SigProfiler software package called "SigProfiler Topography" that allows the annotation of somatic mutational data with "topographic" information, such as regionally variable chromatin features or replication direction. This tool is a useful addition to the already excellent SigProfiler package and will be of great use to the wider community. I only have minor comments regarding the findings presented in the manuscript.

I have few comments to make, only a handful of points relating to the biological examples chosen for demonstrating the utility of SigProfiler Topography:

- The authors state that S17b is depleted at e.g. H3K4me1. However, I wonder if this could be an indirect effect of a general depletion of S17b in open chromatin. This is a general caveat with the way the simulations are done for statistical significance testing. Using a "genome-wide" background means that it will be difficult to separate co-variables. To test whether S17b is depleted for H3Kme1, one would have to use a background that is more representative of the topology of enhancers, yet doesn't contain an enhancer (e.g. immediate flanking regions of the enhancer). While I believe this is out of the scope of this methods paper, it could be something to discuss in the (overall rather short) discussion section.

###Response: We thank the reviewer for the thoughtful and positive review of our manuscript. We greatly appreciate the kind words about SigProfilerTopography and the recognition of its utility for the broader research community.

We also value the reviewer's astute comment regarding the choice of biological examples and the implications of null hypothesis selection, particularly in analyzing the behavior of specific signatures such as SBS17b. Indeed, as noted, the choice of null hypothesis is crucial for determining the regional behavior of mutational signatures. In the original manuscript, our reported results were based on a genome-wide null hypothesis, reflecting the general distribution of mutations across the genome. However, we agree that alternative null hypotheses, such as one based on the specific topology of enhancers without directly overlapping them, could provide additional insights and refined interpretation.

The reviewer's comment presents an excellent opportunity to highlight the flexibility of SigProfilerTopography in testing different and targeted null hypotheses. To demonstrate this capability, we applied the tool to evaluate mutational patterns of SBS1, SBS13, and SBS17 within specific genomic regions using custom-defined BED files. Specifically, we analyzed mutation distributions in CTCF-bound regions and H3K4me1-marked enhancers, as annotated by ENCODE in normal esophageal tissue. To account for local genomic context, we simulated background mutation rates using only the somatic mutations found within the specified regions of interest, including 10-kilobase flanking regions on both the 3' and 5' ends. This approach ensures that the null hypothesis is derived from somatic mutations within the local region rather than the entire genome. We then compared the observed (real) mutations to simulated mutations within both the regions of interest and their flanking regions for signatures SBS1, SBS13, and SBS17b.

Our findings demonstrate that using a local background mutation rate (region of interest plus 10 kb flanking regions) strengthens the observed effect sizes for both SBS1 and SBS17b. When using a genome-wide background mutation rate, SBS1 showed a 0.82-fold and 0.93-fold depletion in CTCF and H3K4me1 regions, respectively. However, when applying a local background mutation rate, SBS1 was more strongly depleted, with 0.35-fold and 0.70-fold reductions in CTCF and H3K4me1 regions, respectively. A similar trend was observed for SBS17b: based on genome-wide background rates, it was enriched 1.74-fold in CTCF regions and depleted 0.86-fold in H3K4me1 regions. When considering the local background mutation rate, the enrichment of SBS17b in CTCF regions increased to 2.79-fold, while depletion in H3K4me1 regions became more pronounced at 0.60-fold. Importantly, no statistically significant differences were detected for any of the flanking regions for either SBS1 or SBS17b, suggesting that the effects are specific to the regions of interest. Lastly, SBS13 showed no significant differences in either the regions of interest or their flanking regions, regardless of the

background mutation rate used. This new set of results have now been summarized and included within the main manuscript and Fig. 2E, with the text provided below.

"The results presented in Fig. 2D are based on a genome-wide null hypothesis that simulates all mutations across the entire genome. However, some of the reported findings, such as, for example, the depletion of SBS17b in H3K4me1-marked enhancers, could be an indirect effect of a general depletion of SBS17b in open chromatin. To assess this possibility, SigProfilerTopography utilizes SigProfilerSimulator [38] which enables testing of different and targeted null hypotheses, allowing for a more refined analysis of regional mutational rates and patterns. To demonstrate this flexibility, we re-evaluated the distribution of SBS1, SBS13, and SBS17b within CTCF-bound regions and H3K4me1-marked enhancers using a local background mutation rate derived by simulating somatic mutations only within the regions of interest and their 10-kilobase flanking regions. This approach ensures that the null hypothesis is derived from somatic mutations within the local region rather than the entire genome.

The results revealed that using a local background strengthened the observed effect sizes for SBS1 and SBS17b (Fig. 2E). Specifically, the depletion of SBS1 in CTCF and H3K4me1-marked regions was more pronounced, shifting from 0.82-fold and 0.93-fold reductions under the genome-wide null hypothesis to 0.35-fold and 0.70-fold reductions under the local background model. Similarly, for SBS17b, the enrichment in CTCF regions increased from 1.74-fold to 2.79-fold, while the depletion in H3K4me1-marked enhancers became more pronounced, decreasing from 0.86-fold to 0.60-fold. Notably, no significant differences were detected for the flanking regions, reinforcing that these effects are specific to the regions of interest. In contrast, SBS13 showed no significant differences regardless of the background mutation rate used. (Fig. 2E) These results highlight the importance of considering local genomic context when analyzing regional mutational patterns and underscore the adaptability of SigProfilerTopography in testing different null hypotheses to refine biological interpretations."

Comment: - I'm wondering whether the S17 patients in the ESCC cohort are in fact misdiagnosed adenocarcinomas? It would be really interesting to know with certainty whether squamous carcinomas do exhibit S17b, which could have implications for the aetiology of this signature

###Response: We appreciate the reviewer's insightful comment regarding the etiology of signature SBS17b. As part of the original study, the ESCC cohort underwent a rigorous pathology review by a centralized IARC/WHO expert panel of seven gastrointestinal pathologists, ensuring high diagnostic accuracy. Each case was independently assessed by at least two pathologists through blinded digital pathology review, with any discrepancies resolved by a third independent evaluation when necessary. We have now clarified this in the methods section of the manuscript:

"All ESCC cases underwent centralized pathology review by an expert panel of seven gastrointestinal pathologists convened by IARC/WHO. Each case was independently evaluated by at least two pathologists through blinded digital pathology assessment, and any discrepancies were adjudicated by a third independent pathologist to ensure accuracy [34]."

To directly address the reviewer's comment, we also double-checked the annotation for all samples with SBS17b (54 out of the 552 ESCC cases) and confirmed that they were consistently

classified as esophageal squamous cell carcinoma. Only six cases required adjudication by a third independent evaluation, which ultimately confirmed their ESCC classification. These six cases exhibited SBS17b, though not at particularly high levels (i.e., they were not outliers in regard to SBS17b mutation burden). While SBS17a/b are typically associated with and highly enriched in esophageal adenocarcinoma, it is very likely that these signatures may also be relevant to a subset of esophageal squamous cell carcinomas.

Comment: - S17b is also strongly replication-timing asymmetrical, but the authors use S2 as their example. Personally, I thought it would have been nice to show the "deep characterisation" of one signature

###Response: We are grateful to the reviewer for this wonderful idea. We have now expanded Figures 3 and 4 to include results on replication-timing, as well as a comparison of genic versus intergenic mutations and transcription/replication asymmetries for SBS17b, in addition to the previously reported results. This addition provides a more in-depth characterization of a single signature in the manuscript while also allowing for a contrast with other mutational signatures.

Comment: - Isn't the "strand-coordinated analysis" going to be highly overlapping with the findings from replication and transcription strand asymmetry? What other types of biological factors would lead to strand-specific mutation accumulation? Maybe the authors could highlight the benefits of looking at strand-coordinated patterns as opposed to replication/transcription direction

###Response: We appreciate the reviewer's question. The purpose of strand-coordinated analysis is to complement transcription and replication strand asymmetry analyses. Strand-coordinated processivity refers to mutations occurring on the same DNA strand as part of a single mutational event, distinct from transcription and replication strand asymmetries, which summarize multiple events across the genome.

A well-studied example is the processivity of APOBEC enzymes, where mutations are often strand-coordinated due to a single catalytic event, independent of transcription or replication processes. Other genomic events, such as double-strand breaks or localized hypermutation, could similarly lead to strand-specific mutation accumulation. SigProfilerTopography provides a summary of these events, revealing which mutagenic processes are likely responsible and offering greater insight into their potential mechanisms. This additional layer of analysis allows to distinguish between mutations from single, strand-coordinated events and those arising from broader transcription or replication-related asymmetries, enhancing our understanding of the underlying mutagenic processes. We have now added text in the manuscript to better explain the rationale behind strand-coordinated analysis. An excerpt is provided below:

"Strand-coordinated analysis provides an additional layer of insight beyond transcription and replication strand asymmetry by identifying mutations that occur on the same DNA strand as part of a single mutational event [58, 59]. Unlike transcription and replication asymmetries, which summarize multiple independent events across the genome, strand-coordinated processivity highlights localized mutation patterns driven by specific mutagenic mechanisms. For instance, APOBEC enzymes induce strand-coordinated mutations through a single catalytic

event, which can be independent of transcription or replication directionality [60, 61]. Similarly, processes such as double-strand breaks and localized hypermutation can result in strand-specific mutation accumulation [58, 59]."

Comment: - The discussion is a bit thin. The authors could discuss that there are potentially different ways to assign signatures to strands in regions? Also, how about interactions between terms, as in replication<>gene expression<>mutation distribution, or replication direction<>transcription direction<>strand bias (as described in my earlier point)? The discussion could also make a bit more of the ESCC analysis, or at least mention that the tool can be used to really understand underlying biology of mutational processes.

###Response: We thank the reviewer for the constructive feedback regarding the discussion section. In response, we have expanded the discussion to address the points raised, including alternative approaches for assigning signatures to strands, potential interactions between biological processes, and an enhanced focus on the ESCC analysis. Additionally, we have clarified the importance of different null hypotheses, highlighting how genome-wide and localized background mutation rates impact the interpretation of mutational patterns, and discussed the required number of mutations to perform a robust topography analysis for capturing different effect sizes. An excerpt of the modified discussion is provided below for the reviewer's perusal.

"SigProfilerTopography is an open-source Python package that allows understanding the interplay between somatic mutagenesis and the structural and topographical features of a genome. The tool can reveal mutational signature-specific tendencies associated with chromatin organization, histone modifications, and transcription factor binding as well as ones affected by cellular processes such as DNA replication and transcription. As we illustrated by applying the tool to 552 whole-genome sequenced ESCCs, SigProfilerTopography simultaneously examines real somatic mutations and simulated mutations, compares their tendencies, and then elucidates the statistically significant differences for each structural and topographical feature of interest. The tool also seamlessly integrates with other SigProfiler tools and leverages them for parts of its computational workflow, including classification of somatic mutations using SigProfilerMatrixGenerator [65], simulating realistic background mutations with SigProfilerSimulator [38], and assigning mutational signatures to each somatic mutation using SigProfilerAssignment [36]. Notably, applying SigProfilerTopography to other cancer types produces results identical with our previously reported pan-cancer topography analyses [6].

Beyond its core functionality, SigProfilerTopography offers flexibility in defining null hypotheses, allowing users to choose between genome-wide and localized mutation backgrounds for statistical comparisons. By default, simulations assume a distribution of mutations across the entire genome, but a more refined null hypothesis can be applied by restricting the background mutation rate to specific genomic regions of interest, including their flanking sequences. This flexibility is particularly relevant when analyzing mutational patterns in regulatory elements, where the influence of chromatin accessibility, transcription factor binding, and histone modifications may be confounded by broader regional mutation rates. As demonstrated in our ESCC analysis, using a localized background strengthens effect sizes and refines biological interpretations of mutational patterns in CTCF-bound regions and H3K4me1-marked enhancers.

One important consideration in using SigProfilerTopography is evaluating strand asymmetries for different mutational signatures. The tool currently infers strand asymmetries by considering replication timing and transcriptional strand orientation separately, yet mutational processes can be influenced by interactions between these factors [19]. Future enhancements could allow for more sophisticated models that jointly consider replication direction, transcription direction, and strand bias, as incorporating these interactions could further refine insights into the mechanistic basis of mutation accumulation.

Additionally, SigProfilerTopography could be expanded to explore even more complex interactions between genomic features. For instance, while the current implementation assesses individual features such as replication timing, chromatin accessibility, and histone modifications independently, future iterations could integrate multi-feature interactions, such as the combined effects of replication timing, gene expression, and mutation distribution. These relationships may be particularly relevant in highly transcribed regions, where differential repair mechanisms and transcription-replication conflicts can influence mutation rates.

Our application of SigProfilerTopography to ESCC provides a compelling example of how the tool can be used to uncover biologically meaningful insights into the underlying processes driving mutagenesis. The ESCC analysis highlighted the differential behavior of SBS1, SBS13, and SBS17b across distinct genomic regions, demonstrating the importance of regional mutational landscapes in shaping mutation distributions. More broadly, the tool facilitates the systematic investigation of how mutational processes interact with the genome topography, chromatin dynamics, and cellular replication and transcription machinery, providing a powerful resource for understanding the etiology of somatic mutations in cancer.

Despite its strengths, SigProfilerTopography has certain limitations. First, the tool can only be used to explore small mutational events including single base substitutions, doublet substitutions, and small insertions and deletions. Currently, the tool does not allow exploring large mutational events [66] such as copy-number changes and structural rearrangements. Second, SigProfilerTopography can be applied only to whole-genome sequenced cancers, and it will not work on whole-exome or targeted cancer gene panel sequencing data as the algorithm requires profiling the non-coding regions of the genome. Lastly, the tool necessitates sufficient numbers of somatic mutations for the statistical analyses to be meaningful and statistically significant. We have previously shown that topographical analyses will work and can yield biologically exciting results when examining adult cancers [6], however, it is currently unclear whether some pediatric cancer genomes will have sufficient numbers of somatic mutations for examining the topography of their mutational signatures. To provide estimates of the necessary number of mutations required to detect different effect sizes, we re-examined our previously published topography analyses [6] to evaluate the minimum statistically detectable effect size and the effect size that is statistically significant in 90% of cases based on the number of mutations assigned to a mutational signature within a cancer type. In this context, statistical significance is defined as a p-value adjusted for multiple hypothesis testing across all signatures within a given cancer type. Our analysis indicates that no effect size could be detected for mutational signatures with fewer than 1,000 mutations in a given cancer type. For signatures with 1,000 to 10,000 mutations, the minimum detectable effect size was a 1.3-fold change, with 90% of detected effect sizes above a 2-fold change. For signatures with 10,000 to 50,000 mutations, the minimum detectable effect size was a 1.2-fold change, with

90% of detected effect sizes above a 1.7-fold change. For signatures with 50,000 to 100,000 mutations, the minimum detectable effect size was a 1.1-fold change, with 90% of detected effect sizes above a 1.5-fold change. For signatures with more than 100,000 mutations, the minimum detectable effect size was a 1.03-fold change, with 90% of detected effect sizes above a 1.2-fold change. These findings emphasize the importance of considering mutation count when designing and interpreting topographic analyses and provide a framework for assessing the applicability of SigProfilerTopography in smaller cohorts or cancer types with low numbers of somatic mutations."

=====
=====

Reviewer #2: Otlu and Alexandrov propose a new Python extension of the SigProfiler suite called SigProfilerTopography, which leverages SigProfilerAssignment and SigProfilerSimulator to assess association of mutation signatures with "topographic" data, including chromatin organisation, DNA replication and transcription, histone modifications and TF binding.

This generic tool is of relevance to the study of mutagenesis, especially the interrelationship between genome topography and mutability/repair. Unlike other existing tools, it simultaneously allows to analyse SBS, DBS and indel signatures in relation to precomputed and user-defined topography data provided in standard formats.

The authors first present the tool and its computational workflow. It works from mutation calls, and signature activities to assign each mutation to a signature. It simulates N (N=100 default) random profiles with similar activities and number of mutation per chromosome to derive a null model against which to compare the observed signal.

It looks at the average observed vs. simulated topography signal in ± 1 kb of the mutation position, for all mutations and for mutations assigned to different signatures.

The authors apply this method to a large published ESCC cohort and find known relationships between signatures and topography signal.

SigProfilerTopography also looks at association of mutations from different signatures with replication timing and replication strand. For this it can leverage Repli-seq data to annotate mutations in early vs. late replicating timing regions, and annotates mutations based on the reference Watson-Crick base pair for assessing association with leading vs. lagging strand mutations. They show that SBS2 (APOBEC) is indeed seen in late-replicating regions and more in the lagging strand, which is compatible with the targeting of single-stranded DNA.

Similarly, to assess whether mutations are related to transcription, SigProfilerTopography can annotate mutations for whether they fall in well-annotated protein coding genes and whether they are on the transcribed strand. They show SBS16, related to alcohol, is seen more in genic regions and on the transcribed strand.

Finally, thanks to the simulated profiles, SigProfiler can detect enrichment in the observed data across the samples for mutations from the same process within 10kb of each other and on the same strand, akin to APOBEC-related kataegis. Interestingly, they find that SBS1, SBS5 and

SBS15 also present many groups of high number of consecutive mutations on the same strand.

In conclusion, SigProfilerTopography is a useful addition to the current SigProfiler suite of tools, which integrates well with the other tools. It deals with SBS, DBS, and indels in whole-genome sequencing data, and works well provided enough mutations as input.

Altogether, I think this is a useful addition to the toolkit for assessing association between mutational signatures and genomewide annotation tracks that could influence mutation rates (here referred to as "topography"). It is definitely worth publication and I believe would be used and well cited.

However, I would have a few comments that I hope could potentially help improve the manuscript and perhaps the tool itself.

###Response: We thank the reviewer for their thorough and thoughtful review of our manuscript. We are pleased to see that the reviewer appreciates the relevance and utility of SigProfilerTopography as a tool for studying the relationship between mutational signatures and genomic topography. We are especially grateful for the reviewer's recognition of the tool's ability to integrate with other SigProfiler tools and its applicability across SBS, DBS, and indel signatures, as well as for their support for publication. We appreciate the constructive comments provided and have carefully considered each one to enhance both the manuscript and the tool.

Comment: Major comments.

1) Software. The software is easy to install and the tutorials are well written. I ran the example and only obtained figures for occupancy, not epigenomics, replication time, strand bias etc. although the logs showed that all analyses ran well. I then realised that while installing the hg19 tracks from the examples, as some of these are relatively large files, some of the files were not downloaded properly. On reinstallation, the software detected that the files were present on disk and did not redownload the corrupt files. I had to delete those files and reinstall. Even after reinstalling those files, I only obtained figures for occupancy.

Here is the code I used after downloading the input data:

```
from SigProfilerTopography import Topography as topography
```

```
topography.install_nucleosome("GRCh37")
topography.install_atac_seq("GRCh37")
topography.install_repli_seq("GRCh37")
```

```
genome = "GRCh37"
inputDir = "21BRCA_vcfs"
outputDir = "results"
jobname = "21BRCA_SPT"
numofSimulations = 5
topography.runAnalyses(genome,
```

```
inputDir,  
outputDir,  
jobname,  
numofSimulations,  
epigenomics=True,  
nucleosome=True,  
replication_time=True,  
strand_bias=True,  
processivity=True)
```

```
jobname = "21BRCA_SPT_with_probability_matrices"  
numofSimulations = 5  
sbs_probability_file =  
"21BRCA_probabilities/COSMIC_SBS96_Decomposed_Mutation_Probabilities.txt"  
dbs_probability_file =  
"21BRCA_probabilities/COSMIC_DBS78_Decomposed_Mutation_Probabilities.txt"
```

```
topography.runAnalyses(genome,  
inputDir,  
outputDir,  
jobname,  
numofSimulations,  
sbs_probabilities = sbs_probability_file,  
dbs_probabilities = dbs_probability_file,  
epigenomics=True,  
nucleosome=True,  
replication_time=True,  
strand_bias=True,  
processivity=True)
```

###Response: We sincerely thank the reviewer for the helpful comments on the software installation and tutorial. Your detailed feedback is greatly appreciated, and we apologize for the issue you encountered.

Based on your report, we were able to reproduce the problem and have since modified the code to address it. We have also conducted multiple tests to ensure that the tool now functions as expected. Specifically, after installation, we intentionally corrupted some of the downloaded files and verified that the software correctly detects and handles the issue upon reinstallation. These modifications should prevent similar problems in the future and improve the overall user experience.

In principle, we strive to extensively test all our code across multiple operating systems and cloud-based instances, including Amazon Web Services (AWS). However, as with any software, occasional issues may arise. As part of the SigProfiler suite, we are fortunate to have a vibrant community of users who actively contribute by submitting GitHub tickets for bug fixes and feature enhancements. We deeply appreciate this collaborative effort and remain committed to continuously refining and improving our tools based on user feedback, including any issues or enhancements related to SigProfilerTopography.

Comment: 2) The analyses of SigProfilerTopography rely on the simulations by SigProfilerSimulator to assess significance, which is a critical part of the analyses. While SigProfilerSimulator is described in another paper, it would be helpful to briefly describe how it works in the methods of this manuscript. In particular, it is not clear to me how the mutations are distributed across the genome during the simulations, i.e. whether it is a uniform coverage only taking into account the nucleotide context and the number of mutations per chromosome or whether some of the topology tracks are also taken into account. Indeed, the simulator could in theory keep the distribution of mutations similar to the observed data in relation to topography tracks (e.g. similar number of mutation in early vs. late replicating regions) for assessing another given topography track (e.g. nucleosome or CTCF occupancy). This could allow to control for interdependencies between the topologies themselves.

###Response: We appreciate the reviewer's insightful suggestion. In response, we have incorporated descriptions of SigProfilerSimulator in both the results and methods sections, emphasizing its capabilities. Below, we provide excerpts from these additions:

Results:

"After processing the input data, SigProfilerTopography simulates all somatic mutations in each sample n times (default of $n=100$) using SigProfilerSimulator [38] (Fig. 1B). SigProfilerSimulator is a computational tool that generates realistic mutational profiles as null hypotheses for statistical testing in cancer genomics. It redistributes mutations across the genome or within specific regions while maintaining key features such as mutational burden across chromosomes (or predefined regions) and local sequence context (e.g., trinucleotide or pentanucleotide context) [38]. By incorporating SigProfilerSimulator, SigProfilerTopography generates biologically constrained simulations to establish an empirical null distribution, enabling the detection of significant mutational enrichments or depletions beyond background noise. By default, the simulation resolution ensures that the total number of mutations per chromosome is preserved, along with the trinucleotide context of each somatic mutation, which includes the mutated base and its immediate 5' and 3' neighboring bases."

Methods:

"SigProfilerTopography leverages SigProfilerSimulator [38] to enable highly customizable simulations by allowing users to specify different mutation resolutions and genomic regions for generating simulated mutations. By default, simulations are distributed across the entire genome while preserving the observed mutation counts for each mutation type and chromosome using the SBS-96, DBS-78, and ID-83 mutation contexts [65]. However, users can refine simulations by matching transcription strand asymmetry to real mutations while maintaining mutation type and chromosome-specific counts. This ensures that other topographic features can be analyzed without altering transcription strand asymmetry. Additionally, simulations can be restricted to specific genomic regions defined by a browser extensible data (BED) file, allowing for targeted investigations of mutational topography."

Additionally, SigProfilerSimulator can be customized to generate null models that align with specific genomic distributions, allowing users to refine hypothesis testing. This flexibility enables the consideration of potential interdependencies between topographic features in analyses. We have clarified this in the manuscript and provided an example demonstrating its

application. Specifically, we show that the enrichment and depletion of mutational signatures in CTCF-bound regions and H3K4me1-marked enhancers can be evaluated using different null hypotheses, such as a genome-wide background or a localized background based on somatic mutations within the regions of interest and their adjacent 10-kilobase flanking regions on both the 3' and 5' ends. An excerpt of the revised manuscript is provided below:

"The results presented in Fig. 2D are based on a genome-wide null hypothesis that simulates all mutations across the entire genome. However, some of the reported findings, such as, for example, the depletion of SBS17b in H3K4me1-marked enhancers, could be an indirect effect of a general depletion of SBS17b in open chromatin. To assess this possibility, SigProfilerTopography utilizes SigProfilerSimulator [38] which enables testing of different and targeted null hypotheses, allowing for a more refined analysis of regional mutational rates and patterns. To demonstrate this flexibility, we re-evaluated the distribution of SBS1, SBS13, and SBS17b within CTCF-bound regions and H3K4me1-marked enhancers using a local background mutation rate derived by simulating somatic mutations only within the regions of interest and their 10-kilobase flanking regions. This approach ensures that the null hypothesis is derived from somatic mutations within the local region rather than the entire genome.

The results revealed that using a local background strengthened the observed effect sizes for SBS1 and SBS17b (Fig. 2E). Specifically, the depletion of SBS1 in CTCF and H3K4me1-marked regions was more pronounced, shifting from 0.82-fold and 0.93-fold reductions under the genome-wide null hypothesis to 0.35-fold and 0.70-fold reductions under the local background model. Similarly, for SBS17b, the enrichment in CTCF regions increased from 1.74-fold to 2.79-fold, while the depletion in H3K4me1-marked enhancers became more pronounced, decreasing from 0.86-fold to 0.60-fold. Notably, no significant differences were detected for the flanking regions, reinforcing that these effects are specific to the regions of interest. In contrast, SBS13 showed no significant differences regardless of the background mutation rate used. (Fig. 2E) These results highlight the importance of considering local genomic context when analyzing regional mutational patterns and underscore the adaptability of SigProfilerTopography in testing different null hypotheses to refine biological interpretations."

Comment: 3) For the mutation-cluster analysis, it seems that the FDR is often low for group length 2 across signatures. Could the authors elaborate why they think that is? Could it be that the simulations are too uniform and the effective genome size, i.e. the cancer-type-specific mutable genome portion, is shorter than the genome portion used for the simulations (see doi.org/10.1038/s41588-021-01005-8) and thus for the same number of mutations, the number of length-2 groups is higher in the observed compared to the simulations? Also related to that, does the simulator take into account "bad genomic regions" where mutation calls are difficult and thus not observed?

###Response: We thank the reviewer for this insightful question. The prevalence of low FDR values for mutation clusters of length 2 across signatures primarily reflects omikli events, which were first described by Mas-Ponte et al. (2020) [PMID: 32747826] and later extended and quantified as the most prevalent type of clustered mutations in human cancers by Bergstrom et al. (2022) [PMID: 35140399].

Additionally, we acknowledge the possibility that the cancer-type-specific mutable genome

portion may be shorter than the genome portion used for the simulations, as suggested by Demeulemeester et al. (2022) [PMID: 35145300]. If the effective mutable genome is more constrained in the observed data compared to simulations, this could lead to an increased number of length-2 mutation groups in real data relative to expectations. We have incorporated this as an additional potential explanation in the manuscript.

Regarding genomic regions where mutation calls are difficult, the current analysis excludes blacklisted regions in the human genome to minimize potential biases. However, as noted by the reviewer, it remains possible that not all regions of the genome are equally mutable, which may further contribute to differences between observed and simulated mutation clustering patterns. We appreciate the reviewer's suggestion, and we have now clarified these points in the revised manuscript. An excerpt of the text is provided below:

"Lastly, we observed an enrichment of mutation clusters of length two across most mutational signatures (Fig. 5D), a previously described phenomenon primarily driven by omikli events [63], a common subclass of clustered mutations in human cancers [58]. However, since the effective mutable genome size may be smaller than the genomic portion used for simulations [64], some of this enrichment could also stem from an overrepresentation in the observed data."

Comment: 4) It is mentioned that it is unclear what the lower bound for the number of mutations would be to get signal and therefore, whether the tool would be applicable to pediatric cancers. Given the known number of mutations in pediatric cancers, simulations or downsampling of the ESCC mutations could help answer this question and this would also be useful information to future users with smaller cohorts.

###Response: We apologize for any lack of clarity in our initial description. In principle, there is no strict lower bound for running SigProfilerTopography; however, the lower the number of mutations, the lower the effect size that can be detected. We also appreciate the reviewer's insightful comment and agree that providing guidance on the number of mutations required for signal detection would be highly beneficial.

To address this concern, we leveraged our previous topography analyses, published in Cell Reports in August 2023, to evaluate the statistically minimum detectable effect size and the statistically significant effect size detectable in 90% of cases based on the number of mutations assigned to a mutational signature within a cancer type. In this context, statistical significance is defined as a p-value adjusted for multiple hypothesis testing across all signatures within a given cancer type. Our analysis indicates that no effect size could be detected for mutational signatures with fewer than 1,000 mutations in a given cancer type. For signatures with 1,000 to 10,000 mutations, the minimum detectable effect size was a 1.3-fold change, with 90% of detected effect sizes above a 2-fold change. For signatures with 10,000 to 50,000 mutations, the minimum detectable effect size was a 1.2-fold change, with 90% of detected effect sizes above a 1.7-fold change. For signatures with 50,000 to 100,000 mutations, the minimum detectable effect size was a 1.1-fold change, with 90% of detected effect sizes above a 1.5-fold change. For signatures with more than 100,000 mutations, the minimum detectable effect size was a 1.03-fold change, with 90% of detected effect sizes above a 1.2-fold change. These results provide guidance to researchers on the effect sizes that can be detected in their data, both for pediatric cancers and other settings with lower mutation

burdens. They offer useful benchmarks for assessing the applicability of our tool to smaller cohorts. To reflect this, we have revised the discussion section of the manuscript to incorporate these findings. An excerpt of the modified discussion is provided below:

"To provide estimates of the necessary number of mutations required to detect different effect sizes, we re-examined our previously published topography analyses [6] to evaluate the minimum statistically detectable effect size and the effect size that is statistically significant in 90% of cases based on the number of mutations assigned to a mutational signature within a cancer type. In this context, statistical significance is defined as a p-value adjusted for multiple hypothesis testing across all signatures within a given cancer type. Our analysis indicates that no effect size could be detected for mutational signatures with fewer than 1,000 mutations in a given cancer type. For signatures with 1,000 to 10,000 mutations, the minimum detectable effect size was a 1.3-fold change, with 90% of detected effect sizes above a 2-fold change. For signatures with 10,000 to 50,000 mutations, the minimum detectable effect size was a 1.2-fold change, with 90% of detected effect sizes above a 1.7-fold change. For signatures with 50,000 to 100,000 mutations, the minimum detectable effect size was a 1.1-fold change, with 90% of detected effect sizes above a 1.5-fold change. For signatures with more than 100,000 mutations, the minimum detectable effect size was a 1.03-fold change, with 90% of detected effect sizes above a 1.2-fold change. These findings emphasize the importance of considering mutation count when designing and interpreting topographic analyses and provide a framework for assessing the applicability of SigProfilerTopography in smaller cohorts or cancer types with low numbers of somatic mutations."

Comment: Minor comment.

1) Figure 3B is referenced later than panels other than 3A; maybe change the order?

###Response: Thank you for catching this discrepancy. We have restructured the text to ensure that the panels in Figure 3 are referenced in the correct order.

=====
=====

Reviewer #3: Reviewer's comments

The authors are recognized as one of the leading research teams in the field of mutational signatures. The manuscript they have submitted appears to extend the analyses presented in their recent Cell Reports publication (2023) into a software paper for Genome Biology, with the intention of making it accessible to the broader cancer mutation research community.

###Response: Thank you very much for your kind words! They are greatly appreciated by our team.

Comment: Regrettably, I find that the manuscript, in its current form, does not meet the standards required for publication in the Software section of Genome Biology for the following reasons:

- The software presented lacks sufficient novelty. It would be more appropriate to offer it as part

of the supplementary materials in their Cell Reports (2023) publication, where it can still be of value to the global research community.

###Response: We appreciate the reviewer's candid concern regarding novelty and welcome the opportunity to clarify the distinct contributions of our current manuscript compared to our previous publication in Cell Reports (2023).

The Cell Reports paper focused on characterizing the topography of mutational signatures in human cancer, providing foundational biological insights. In contrast, this manuscript introduces SigProfilerTopography, a robust and user-friendly software tool that allows researchers to independently perform these analyses without the need for custom and complex computational workflows. By making these methods widely accessible, this work enhances usability and facilitates broader adoption of topographic mutational signature analyses.

In our previous Cell Reports study, we analyzed the topography of over 70 mutational signatures across 5,120 whole-genome-sequenced tumors from 40 cancer types, making the results publicly available as part of the COSMIC database. Here, we introduce SigProfilerTopography and demonstrate its capabilities by analyzing mutational signatures in esophageal squamous cell carcinoma (ESCC). Additionally, based on another reviewer's feedback, we now highlight the tool's flexibility in testing different null hypotheses. Specifically, we evaluate mutational patterns of SBS1, SBS13, and SBS17b within CTCF-bound regions and H3K4me1-marked enhancers in ESCC using both a genome-wide background and a localized background derived from somatic mutations within the regions of interest and their adjacent 10-kilobase flanking regions. By comparing observed and simulated mutation distributions, this analysis demonstrates the tool's ability to refine mutational context assessments and test targeted hypotheses.

Specifically, our findings demonstrate that using a local background mutation rate (region of interest plus 10 kb flanking regions) strengthens the observed effect sizes for both SBS1 and SBS17b. When using a genome-wide background mutation rate, SBS1 showed a 0.82-fold and 0.93-fold depletion in CTCF and H3K4me1 regions, respectively. However, when applying a local background mutation rate, SBS1 was more strongly depleted, with 0.35-fold and 0.70-fold reductions in CTCF and H3K4me1 regions, respectively. A similar trend was observed for SBS17b: based on genome-wide background rates, it was enriched 1.74-fold in CTCF regions and depleted 0.86-fold in H3K4me1 regions. When considering the local background mutation rate, the enrichment of SBS17b in CTCF regions increased to 2.79-fold, while depletion in H3K4me1 regions became more pronounced at 0.60-fold. Importantly, no statistically significant differences were detected for any of the flanking regions for either SBS1 or SBS17b, suggesting that the effects are specific to the regions of interest. Lastly, SBS13 showed no significant differences in either the regions of interest or their flanking regions, regardless of the background mutation rate used. This new set of results have now been summarized and included within the main manuscript and Fig. 2E, with the text provided below.

"The results presented in Fig. 2D are based on a genome-wide null hypothesis that simulates all mutations across the entire genome. However, some of the reported findings, such as, for example, the depletion of SBS17b in H3K4me1-marked enhancers, could be an indirect effect of a general depletion of SBS17b in open chromatin. To assess this possibility,

SigProfilerTopography utilizes SigProfilerSimulator [38] which enables testing of different and targeted null hypotheses, allowing for a more refined analysis of regional mutational rates and patterns. To demonstrate this flexibility, we re-evaluated the distribution of SBS1, SBS13, and SBS17b within CTCF-bound regions and H3K4me1-marked enhancers using a local background mutation rate derived by simulating somatic mutations only within the regions of interest and their 10-kilobase flanking regions. This approach ensures that the null hypothesis is derived from somatic mutations within the local region rather than the entire genome.

The results revealed that using a local background strengthened the observed effect sizes for SBS1 and SBS17b (Fig. 2E). Specifically, the depletion of SBS1 in CTCF and H3K4me1-marked regions was more pronounced, shifting from 0.82-fold and 0.93-fold reductions under the genome-wide null hypothesis to 0.35-fold and 0.70-fold reductions under the local background model. Similarly, for SBS17b, the enrichment in CTCF regions increased from 1.74-fold to 2.79-fold, while the depletion in H3K4me1-marked enhancers became more pronounced, decreasing from 0.86-fold to 0.60-fold. Notably, no significant differences were detected for the flanking regions, reinforcing that these effects are specific to the regions of interest. In contrast, SBS13 showed no significant differences regardless of the background mutation rate used. (Fig. 2E) These results highlight the importance of considering local genomic context when analyzing regional mutational patterns and underscore the adaptability of SigProfilerTopography in testing different null hypotheses to refine biological interpretations."

Comment: - While the authors have provided an overview of the software on their website, the manuscript lacks the necessary depth and structure. To enhance its clarity for the readership, it should include a more detailed description of the software's architecture, implementation, and the methodologies employed (including statistical considerations).

###Response: We appreciate the reviewer's feedback and recognize the importance of providing greater clarity on the software's architecture, implementation, and methodologies within the manuscript. In response, we have expanded the manuscript's methods section to include a more detailed description of these aspects while adhering to the journal's formatting limitations. This revised section is provided below, and it now offers an overview of the software's structure, key implementation details, and the statistical methodologies used, along with their relevant considerations. We hope these additions improve the clarity and depth of the manuscript, making it more informative and accessible to readers.

"SigProfilerTopography is developed as a computationally efficient Python package, and it is available for installation through PyPI. The tool leverages SigProfilerAssignment for attributing mutational signatures to individual somatic mutations [36], SigProfilerSimulator for generating all simulated datasets [38], and SigProfilerMatrixGenerator for processing input data for somatic mutations [65]. SigProfilerTopography leverages SigProfilerSimulator [38] to enable highly customizable simulations by allowing users to specify different mutation resolutions and genomic regions for generating simulated mutations. By default, simulations are distributed across the entire genome while preserving the observed mutation counts for each mutation type and chromosome using the SBS-96, DBS-78, and ID-83 mutation contexts [65]. However, users can refine simulations by matching transcription strand asymmetry to real mutations while maintaining mutation type and chromosome-specific counts. This ensures that other topographic features can be analyzed without altering transcription strand asymmetry.

Additionally, simulations can be restricted to specific genomic regions defined by a browser extensible data (BED) file, allowing for targeted investigations of mutational topography. SigProfilerTopography allows processing all types of small mutational events, including: (i) single base substitutions, (ii) doublet base substitutions, and (iii) small insertions and deletions. The tool supports most commonly used data formats for somatic mutations: Variant Calling Format (VCF), Mutation Annotation Format (MAF), International Cancer Genome Consortium (ICGC) data format, and simple text file. SigProfilerTopography allows examining topography features in wiggle (wig), browser extensible data (bed), bigWig, and bigBed formats. The tool has been extensively tested on data from transposase-accessible chromatin with sequencing (ATAC-Seq), replication sequencing (Repli-Seq), micrococcal nuclease sequencing (MNase-Seq), and immunoprecipitation sequencing (ChIP-Seq). As shown in the manuscript, these data can be analyzed to evaluate occupancy, transcription strand asymmetry, replication strand asymmetry, replication timing, and strand-coordinated mutagenesis. The methodology of each of these analyses is described in the subsequent sections.

Occupancy Analysis: For each real somatic mutation, SigProfilerTopography accumulates both the total signal intensity (SUM) and the number of signals (COUNT). The average signal at each base is then calculated as SUM/COUNT. This process is repeated for simulated mutations, allowing for statistical comparison between observed and simulated data. Confidence intervals around the average simulated occupancy signals are generated, and significance is assessed using a z-test. The p-values are combined using Fisher's method and corrected for multiple testing using the Benjamini-Hochberg procedure.

Replication Timing: Replication timing signals are sorted in descending order and divided into deciles, each containing 10% of the data. The first decile represents the earliest replicating region, while the last decile corresponds to the latest replicating region. Real somatic mutations are assigned to deciles based on their overlap with replication domains, and mutation densities are calculated by normalizing against the number of valid bases (A, T, G, C) in each decile. Densities are further normalized relative to the highest mutation density. This process is repeated for simulated mutations, enabling the generation of confidence intervals around the normalized mutation densities.

Strand Asymmetry: In the Repli-seq signal data, local maxima correspond to replication initiation zones (peaks), while local minima represent replication termination zones (valleys). Peaks and valleys are sorted by genomic coordinates, and regions of at least 10 kb with a positive slope are annotated as leading strands, while negative slopes define lagging strands. To ensure robustness, the last 25 kb of replication termination zones are excluded. Fisher's exact test is used to compare the observed versus simulated mutation counts on each strand, with multiple testing correction via the Benjamini-Hochberg method. Statistically significant strand asymmetries (adjusted $p \leq 0.05$) are reported, with odds ratios above 1.10 used to identify enriched mutation biases across lagging versus leading, transcribed versus untranscribed, and genic versus intergenic regions.

Strand-Coordinated Mutagenesis: For each SBS signature and group length, the observed number of clustered mutation groups is compared to the expected number derived from 100 simulated datasets, serving as the null hypothesis. Statistical significance is assessed using z-tests, determining whether the observed mean differs significantly from the simulated mean. The p-values are adjusted for multiple testing using the Benjamini-Hochberg method, and only

SBS signatures and group lengths with an adjusted $p \leq 0.05$ are considered statistically significant and reported.

Statistical Testing: By default, the tool performs statistical comparisons and Benjamini-Hochberg corrections for multiple hypothesis testing using the statsmodels Python package. Where appropriate, Fisher's method is used to combine p-values. Adjusted p-values are considered statistically significant if they are below 0.05.

Tool Availability: SigProfilerTopography is freely available, distributed under the BSD-2-Clause license, and has been extensively documented.

Python code: <https://github.com/AlexandrovLab/SigProfilerTopography>

Documentation: <https://osf.io/5unby/wiki/home/>

Comment: - The authors mention three known limitations in the Discussion section. Addressing these limitations in the software's implementation would significantly elevate SigProfilerTopography to a level that could meet the criteria for publication in Genome Biology.

###Response: We thank the reviewer for their thoughtful suggestion. After discussing this feedback with the editorial team, we recognize that while addressing these limitations could indeed enhance the functionality of SigProfilerTopography, implementing these changes is beyond the scope of the current study. Our primary aim here is to provide a streamlined and accessible tool that performs core topography analyses efficiently, as outlined in the manuscript.

We hope the reviewer will appreciate the value of this tool in its current form, designed to facilitate broad application across the research community. Future work could certainly explore incorporating these advanced features as the tool evolves. Lastly, it should be noted that we have substantially expanded the discussion of the manuscript to further explain SigProfilerTopography as well as its utility and limitations.

Comment: - Additionally, it is crucial to demonstrate whether the software can achieve a comparable performance across pan-cancer analyses, as seen in the analysis of the ESCC cancer type presented in the manuscript.

In conclusion, the manuscript, in its current form, falls short of the expectations for a software paper in Genome Biology.

###Response: We thank the reviewer for highlighting the importance of demonstrating pan-cancer performance. In response, we have clarified in the discussion section that the software can indeed achieve comparable performance across pan-cancer analyses, as evidenced by results referenced in our previous Cell Reports publication. We believe this additional information reinforces the robustness and applicability of the software beyond the ESCC cancer type presented within the manuscript.

We hope these revisions address the reviewer's concerns and provide greater confidence in the software's performance and relevance across diverse cancer types.

Second round of review

Reviewer 1

The authors have addressed all my concerns.

Reviewer 2

I thank the authors for their thorough responses to my comments. In particular, the local 10 kb background is ingenious and should naturally help control for different local technical and biological factors. I have no further questions.